# Batch Bayesian Optimization for Replicable Experimental Design

**Zhongxiang Dai**[1], **Quoc Phong Nguyen**[2], **Sebastian Shenghong Tay**[1,4],
**Daisuke Urano**[5], **Richalynn Leong**[5], **Bryan Kian Hsiang Low**[1], **Patrick Jaillet**[2,3]
[1]Department of Computer Science, National University of Singapore
[2]LIDS and [3]EECS, Massachusetts Institute of Technology
[4]Institute for Infocomm Research (I2R), A*STAR, Singapore
[5]Temasek Life Sciences Laboratory, Singapore
dzx@nus.edu.sg, qphongmp@gmail.com, sebastian.tay@u.nus.edu,
{daisuke, richalynn}@tll.org.sg, lowkh@comp.nus.edu.sg, jaillet@mit.edu

## Abstract

Many real-world experimental design problems *(a)* evaluate multiple experimental conditions in parallel and *(b)* replicate each condition multiple times due to large and heteroscedastic observation noise. Given a fixed total budget, this naturally induces a trade-off between *evaluating more unique conditions while replicating each of them fewer times* vs. *evaluating fewer unique conditions and replicating each more times*. Moreover, in these problems, practitioners may be risk-averse and hence prefer an input with both good average performance and small variability. To tackle both challenges, we propose the *Batch Thompson Sampling for Replicable Experimental Design* (BTS-RED) framework, which encompasses three algorithms. Our BTS-RED-Known and BTS-RED-Unknown algorithms, for, respectively, known and unknown noise variance, choose the number of replications *adaptively* rather than deterministically such that an input with a larger noise variance is replicated more times. As a result, despite the noise heteroscedasticity, both algorithms enjoy a theoretical guarantee and are *asymptotically no-regret*. Our Mean-Var-BTS-RED algorithm aims at risk-averse optimization and is also asymptotically no-regret. We also show the effectiveness of our algorithms in two practical real-world applications: precision agriculture and AutoML.

## 1 Introduction

*Bayesian optimization* (BO), which is a sequential algorithm for optimizing black-box and expensive-to-evaluate functions [14, 15], has found application in a wide range of experimental design problems [21]. Many such applications which use BO to accelerate the scientific discovery process [27] fall under the umbrella of AI for science (*AI4Science*). Many real-world experimental design problems, such as precision agriculture, share two inherent characteristics: *(a)* multiple experimental conditions are usually evaluated in parallel to take full advantage of the available experimental budget; *(b)* the evaluation of every experimental condition is usually *replicated* multiple times [31] because every experiment may be associated with a large and heteroscedastic (i.e., input-dependent) observation noise, in which case replication usually leads to better performances [2, 33, 43]. Replicating each evaluated experimental condition is also a natural choice in experimental design problems in which it incurs considerable setup costs to test every new experimental condition. This naturally induces an interesting challenge regarding the trade-off between input selection and replication: in every iteration of BO where we are given a fixed total experimental budget, should we *evaluate more unique experimental conditions and replicate each of them fewer times* or *evaluate fewer unique conditions and replicate each more times*? Interestingly, this trade-off is also commonly found in

other applications such as automated machine learning (AutoML), in which parallel evaluations are often adopted to exploit all available resources [28] and heteroscedasticity is a prevalent issue [12]. Furthermore, these experimental design problems with large and *heteroscedastic noise* are often faced with another recurring challenge: instead of an input experimental condition (e.g., a hyperparameter configuration for an ML model) that produces a good performance (e.g., large validation accuracy) on average, some practitioners may be *risk-averse* and instead prefer an input that both yields a good average performance and has small variability. As a result, instead of only maximizing the mean of the black-box function, these risk-averse practitioners may instead look for inputs with both a large mean function value and a small noise variance [26, 33].

In this work, we provide solutions to both challenges in a principled way by proposing the framework of *Batch Thompson Sampling for Replicable Experimental Design* (BTS-RED). The first challenge regarding the trade-off between input selection and replication is tackled by the first two incarnations of our framework: the BTS-RED-Known (Sec. 3.1) and BTS-RED-Unknown (Sec. 3.2) algorithms, which are applicable to scenarios where the noise variance function is known or unknown, respectively. For batch selection, we adopt the Thompson sampling (TS) strategy because its inherent randomness makes it particularly simple to select a batch of inputs [28]. Moreover, previous works on BO have shown that the use of TS both allows for the derivation of theoretical guarantees [16, 28] and leads to strong empirical performances [16, 20]. For replication selection, instead of the common practice of replicating every queried input a fixed number of times, we *choose the number of replications adaptively depending on the observation noise*. Specifically, in every iteration, both algorithms repeat the following two steps until the total budget is exhausted: *(a)* choose an input query following the TS strategy, and then *(b)* adaptively choose the number of replications for the selected input such that *an input with a larger noise variance is replicated more times*. Of note, in spite of the noise heteroscedasticity, our principled approach to choosing the number of replications ensures that the *effective noise variance* $R^2$ of every queried input is the same (Sec. 3.1). This allows us to derive an upper bound on their cumulative regret and show that they are *asymptotically no-regret*. Our theoretical guarantee formalizes the impact of the properties of the experiments, i.e., our regret upper bound becomes better if the total budget is increased or if the overall noise level is reduced. Importantly, our theoretical result provides a guideline on the choice of the effective noise variance parameter $R^2$, which is achieved by minimizing the regret upper bound and allows $R^2$ to automatically adapt to the budgets and noise levels of different experiments (Sec. 3.1.2).

To handle the second challenge of risk-averse optimization, we propose the third variant of our BTS-RED framework named Mean-Var-BTS-RED (Sec. 4), which is a natural extension of BTS-RED-Unknown. Mean-Var-BTS-RED aims to maximize the mean-variance objective function, which is a weighted combination of the mean objective function and negative noise variance function (Sec. 2). We prove an upper bound on the mean-variance cumulative regret of Mean-Var-BTS-RED (Sec. 4) and show that it is also *asymptotically no-regret*.

In addition to our theoretical contributions, we also demonstrate the practical efficacy of our algorithms in two real-world problems (Sec. 5). Firstly, in real-world precision agriculture experiments, plant biologists usually *(a)* evaluate multiple growing conditions in parallel, and *(b)* replicate each condition multiple times to get a reliable outcome [31]. Moreover, plant biologists often prefer more replicable conditions, i.e., inputs with small noise variances. This is hence an ideal application for our algorithms. So, we conduct an experiment using real-world data on plant growths, to show the effectiveness of our algorithms in precision agriculture (Sec. 5.2). Next, we also apply our algorithms to AutoML to find hyperparameter configurations with competitive and *reproducible* results across different AutoML tasks (Sec. 5.3). The efficacy of our algorithms demonstrates their capability to improve the reproducibility of AutoML tasks which is an important issue in AutoML [25].

## 2   Background

We denote by $f : \mathcal{X} \to \mathbb{R}$ the objective function we wish to maximize, and by $\sigma^2 : \mathcal{X} \to \mathbb{R}^+$ the input-dependent noise variance function. We denote the minimum and maximum noise variance as $\sigma_{\min}^2$ and $\sigma_{\max}^2$. For simplicity, we assume that the domain $\mathcal{X}$ is finite, since extension to compact domains can be easily achieved via suitable discretizations [5]. After querying an input $\boldsymbol{x} \in \mathcal{X}$, we observe a noisy output $y = f(\boldsymbol{x}) + \epsilon$ where $\epsilon \sim \mathcal{N}(0, \sigma^2(\boldsymbol{x}))$. In every iteration $t$, we select a batch of $b_t \geq 1$ inputs $\{\boldsymbol{x}_t^{(b)}\}_{b=1,\ldots,b_t}$, and query every $\boldsymbol{x}_t^{(b)}$ with $n_t^{(b)} \geq 1$ parallel processes. We denote the *total budget* as $\mathbb{B}$ such that $\sum_{b=1}^{b_t} n_t^{(b)} \leq \mathbb{B}, \forall t \geq 1$. We model the function $f$

using a *Gaussian process* (GP) [40]: $\mathcal{GP}(\mu(\cdot), k(\cdot, \cdot))$, where $\mu(\cdot)$ is a mean function which we assume w.l.o.g. $\mu(\boldsymbol{x}) = 0$ and $k(\cdot, \cdot)$ is a kernel function for which we focus on the commonly used *squared exponential* (SE) kernel. In iteration $t$, we use the observation history in the first $t-1$ iterations (batches) to calculate the GP posterior $\mathcal{GP}(\mu_{t-1}(\cdot), \sigma^2_{t-1}(\cdot, \cdot))$, in which $\mu_{t-1}(\cdot)$ and $\sigma^2_{t-1}(\cdot, \cdot)$ represent the GP posterior mean and covariance functions (details in Appendix A). For BTS-RED-Unknown and Mean-Var-BTS-RED (i.e., when $\sigma^2(\cdot)$ is unknown), we use another GP, denoted as $\mathcal{GP}'$, to model $-\sigma^2(\cdot)$ (Sec. 3.2), and denote its posterior as $\mathcal{GP}'(\mu'_{t-1}(\cdot), \sigma'^2_{t-1}(\cdot, \cdot))$.

In our theoretical analysis of BTS-RED-Known and BTS-RED-Unknown where we aim to maximize $f$, we follow previous works on batch BO [11, 18, 35] and derive an upper bound on the *batch cumulative regret* $R_T = \sum_{t=1}^{T} \min_{b \in [b_t]} [f(\boldsymbol{x}^*) - f(\boldsymbol{x}_t^{(b)})]$, in which $\boldsymbol{x}^* \in \arg\max_{\boldsymbol{x} \in \mathcal{X}} f(\boldsymbol{x})$ and we have used $[b_t]$ to denote $\{1, \ldots, b_t\}$. We show (Sec. 3) that both BTS-RED-Known and BTS-RED-Unknown enjoy a sub-linear upper bound on $R_T$, which suggests that as $T$ increases, a global optimum $\boldsymbol{x}^*$ is guaranteed to be queried since the *batch simple regret* $S_T = \min_{t \in [T]} \min_{b \in [b_t]} [f(\boldsymbol{x}^*) - f(\boldsymbol{x}_t^{(b)})] \leq R_T/T$ goes to $0$ asymptotically. We analyze the batch cumulative regret because it allows us to show the benefit of batch evaluations, and our analysis can also be modified to give an upper on the sequential cumulative regret of $R'_T = \sum_{t=1}^{T} \sum_{b=1}^{b_t} [f(\boldsymbol{x}^*) - f(\boldsymbol{x}_t^{(b)})]$ (Appendix D). Our Mean-Var-BTS-RED aims to maximize the *mean-variance objective function* $h_\omega(\boldsymbol{x}) = \omega f(\boldsymbol{x}) - (1 - \omega)\sigma^2(\boldsymbol{x}) = \omega f(\boldsymbol{x}) + (1 - \omega)g(\boldsymbol{x})$, in which we have defined $g(\boldsymbol{x}) \triangleq -\sigma^2(\boldsymbol{x}), \forall \boldsymbol{x} \in \mathcal{X}$. The user-specified weight parameter $\omega \in [0, 1]$ reflects our relative preference for larger mean function values or smaller noise variances. Define the *mean-variance batch cumulative regret* as $R_T^{\mathrm{MV}} = \sum_{t=1}^{T} \min_{b \in [b_t]} [h_\omega(\boldsymbol{x}_\omega^*) - h_\omega(\boldsymbol{x}_t^{(b)})]$ where $\boldsymbol{x}_\omega^* \in \arg\max_{\boldsymbol{x} \in \mathcal{X}} h_\omega(\boldsymbol{x})$. We also prove a sub-linear upper bound on $R_T^{\mathrm{MV}}$ for Mean-Var-BTS-RED (Sec. 4).

## 3 BTS-RED-Known and BTS-RED-Unknown

Here, we firstly introduce BTS-RED-Known and its theoretical guarantees (Sec. 3.1), and then discuss how it can be extended to derive BTS-RED-Unknown (Sec. 3.2).

### 3.1 BTS-RED with Known Noise Variance Function

#### 3.1.1 BTS-RED-Known

---
**Algorithm 1** BTS-RED-Known.

---
1: **for** $t = 1, 2, \ldots, T$ **do**
2:     $b = 0, n_t^{(0)} = 0$
3:     **while** $\sum_{b'=0}^{b} n_t^{(b')} < \mathbb{B}$ **do**
4:        $b \leftarrow b + 1$
5:        Sample a function $f_t^{(b)}$ from the GP posterior of $\mathcal{GP}(\mu_{t-1}(\cdot), \beta_t^2 \sigma^2_{t-1}(\cdot, \cdot))$ (Sec. 2)
6:        Choose $\boldsymbol{x}_t^{(b)} = \arg\max_{\boldsymbol{x} \in \mathcal{X}} f_t^{(b)}(\boldsymbol{x})$ and $n_t^{(b)} = \lceil \sigma^2(\boldsymbol{x}_t^{(b)})/R^2 \rceil$
7:     $b_t = b - 1$
8:     **for** $b \in [b_t]$, query $\boldsymbol{x}_t^{(b)}$ with $n_t^{(b)}$ parallel processes
9:     **for** $b \in [b_t]$, observe $\{y_{t,n}^{(b)}\}_{n \in [n_t^{(b)}]}$. Calculate their empirical mean $y_t^{(b)} = (1/n_t^{(b)}) \sum_{n=1}^{n_t^{(b)}} y_{t,n}^{(b)}$
10:    Use $\{(\boldsymbol{x}_t^{(b)}, y_t^{(b)})\}_{b \in [b_t]}$ to update posterior of $\mathcal{GP}$

---

BTS-RED-Known (Algo. 1) assumes that $\sigma^2(\cdot)$ is known. In every iteration $t$, to sequentially select every $\boldsymbol{x}_t^{(b)}$ and its corresponding $n_t^{(b)}$, we repeat the following process until the total number of replications has consumed the total budget $\mathbb{B}$ (i.e., until $\sum_{b'=1}^{b} n_t^{(b')} \geq \mathbb{B}$, line 3 of Algo. 1):

- **line 5**: sample a function $f_t^{(b)}$ from $\mathcal{GP}(\mu_{t-1}(\cdot), \beta_t^2 \sigma^2_{t-1}(\cdot, \cdot))$ ($\beta_t$ will be defined in Theorem 3.1);
- **line 6**: choose $\boldsymbol{x}_t^{(b)} = \arg\max_{\boldsymbol{x} \in \mathcal{X}} f_t^{(b)}(\boldsymbol{x})$ by maximizing the sampled function $f_t^{(b)}$, and choose $n_t^{(b)} = \lceil \sigma^2(\boldsymbol{x}_t^{(b)})/R^2 \rceil$, where $\lceil \cdot \rceil$ is the ceiling operator and $R^2$ is the *effective noise variance*.

After the entire batch of $b_t$ inputs have been selected, every $\boldsymbol{x}_t^{(b)}$ is queried with $n_t^{(b)}$ parallel processes (line 8), and the empirical mean $y_t^{(b)}$ of these $n_t^{(b)}$ observations is calculated (line 9).

Finally, $\{(\boldsymbol{x}_t^{(b)}, y_t^{(b)})\}_{b \in [b_t]}$ are used to update the posterior of $\mathcal{GP}$ (line 10). Of note, since the observation noise is assumed to be Gaussian-distributed with a variance of $\sigma^2(\boldsymbol{x}_t^{(b)})$ (Sec. 2), after querying $\boldsymbol{x}_t^{(b)}$ independently for $n_t^{(b)}$ times, the empirical mean $y_t^{(b)}$ follows a Gaussian distribution with noise variance $\sigma^2(\boldsymbol{x}_t^{(b)})/n_t^{(b)}$. Next, since we select $n_t^{(b)}$ by $n_t^{(b)} = \lceil \sigma^2(\boldsymbol{x}_t^{(b)})/R^2 \rceil$ (line 6), $\sigma^2(\boldsymbol{x}_t^{(b)})/n_t^{(b)}$ is guaranteed to be upper-bounded by $R^2$. In other words, *every observed empirical mean $y_t^{(b)}$ follows a Gaussian distribution with a noise variance that is upper-bounded by the effective noise variance $R^2$*. This is crucial for our theoretical analysis since it ensures that the effective noise variance is $R$-sub-Gaussian and thus preserves the validity of the GP-based confidence bound [8].

In practice, since our BTS-RED-Known algorithm only aims to maximize the objective function $f$ (i.e., we are not concerned about learning the noise variance function), some replications may be wasted on undesirable input queries (i.e., those with small values of $f(\boldsymbol{x})$) especially in the initial stage when our algorithm favours exploration. To take this into account, we adopt a simple heuristic: we impose a maximum number of replications denoted as $n_{\max}$, and set $n_{\max} = \mathbb{B}/2$ in the first $T/2$ iterations and $n_{\max} = \mathbb{B}$ afterwards. This corresponds to favouring exploration of more inputs (each with less replications) initially and preferring exploitation in later stages. This technique is also used for BTS-RED-Unknown (Sec. 3.2) yet not adopted for Mean-Var-BTS-RED (Sec. 4) since in mean-variance optimization, we also aim to learn (and minimize) the noise variance function.

Due to our stopping criterion for batch selection (line 3), in practice, some budgets may be unused in an iteration. E.g., when $\mathbb{B} = 50$, if $\sum_{b'=1}^{b-1} n_t^{(b')} = 43$ after the first $b - 1$ selected queries and the newly selected $n_t$ for the $b^{\text{th}}$ query $\boldsymbol{x}_t^{(b)}$ is $n_t^{(b)} = 12$, then the termination criterion is met (i.e., $\sum_{b'=1}^{b} n_t^{(b')} \geq \mathbb{B}$) and only $43/50$ of the budgets are used. So, we adopt a simple technique: in the example above, we firstly evaluate the last selected $\boldsymbol{x}_t^{(b)}$ for 7 times, and in the next iteration $t + 1$, we start by completing the unfinished evaluation of $\boldsymbol{x}_t^{(b)}$ by allocating $12 - 7 = 5$ replications to $\boldsymbol{x}_t^{(b)}$. Next, we run iteration $t + 1$ with the remaining budget, i.e., we let $\mathbb{B} = 50 - 5 = 45$ in iteration $t + 1$.

### 3.1.2 Theoretical Analysis of BTS-RED-Known

Following the common practice in BO [8], we assume $f$ lies in a *reproducing kernel Hilbert space* (RKHS) induced by an SE kernel $k$: $\|f\|_{\mathcal{H}_k} \leq B$ for some $B > 0$ where $\|\cdot\|_{\mathcal{H}_k}$ denotes the RKHS norm. Theorem 3.1 below gives a regret upper bound of BTS-RED-Known (proof in Appendix B).

**Theorem 3.1** (BTS-RED-Known). *Choose $\delta \in (0,1)$. Define $\tau_{t-1} \triangleq \sum_{t'=1}^{t-1} b_{t'}$, and define $\beta_t \triangleq B + R\sqrt{2(\Gamma_{\tau_{t-1}} + 1 + \log(2/\delta))}$ where $\Gamma_{\tau_{t-1}}$ denotes the maximum information gain about $f$ from any $\tau_{t-1}$ observations. With probability of at least $1 - \delta$ ($\widetilde{\mathcal{O}}$ ignores all log factors),*

$$R_T = \widetilde{\mathcal{O}}\left(e^C \sqrt{R^2 / \left(\mathbb{B}/\lceil \frac{\sigma_{\max}^2}{R^2} \rceil - 1\right)} \sqrt{T\Gamma_{T\mathbb{B}}} (\sqrt{C} + \sqrt{\Gamma_{T\mathbb{B}}})\right).$$

*$C$ is a constant s.t. $\max_{A \subset \mathcal{X}, |A| \leq \mathbb{B}} \mathbb{I}\left(f; \boldsymbol{y}_A | \boldsymbol{y}_{1:t-1}\right) \leq C, \forall t \geq 1$. $\mathbb{I}\left(f; \boldsymbol{y}_A | \boldsymbol{y}_{1:t-1}\right)$ is the information gain from observations $\boldsymbol{y}_A$ at inputs $A$, given observations $\boldsymbol{y}_{1:t-1}$ in the first $t - 1$ iterations.*

It has been shown by [19] that by running uncertainty sampling (i.e., choosing the initial inputs by sequentially maximizing the GP posterior variance) as the initialization phase for a finite number (independent of $T$) of iterations, $C$ can be chosen to be a constant independent of $\mathbb{B}$ and $T$. As a result, the regret upper bound from Theorem 3.1 can be simplified into $R_T = \widetilde{\mathcal{O}}\left(\sqrt{R^2/(\mathbb{B}/\lceil \frac{\sigma_{\max}^2}{R^2} \rceil - 1)} \sqrt{T\Gamma_{T\mathbb{B}}}(1 + \sqrt{\Gamma_{T\mathbb{B}}})\right)$. Therefore, for the SE kernel for which $\Gamma_{T\mathbb{B}} = \mathcal{O}(\log^{d+1}(T\mathbb{B}))$, our regret upper bound is sub-linear, which indicates that our BTS-RED-Known is *asymptotically no-regret*. Moreover, the benefit of a larger total budget $\mathbb{B}$ is also reflected from our regret upper bound since it depends on the total budget $\mathbb{B}$ via $\widetilde{\mathcal{O}}((\log^{d+1}(T\mathbb{B}) + \log^{(d+1)/2}(T\mathbb{B}))/\sqrt{\mathbb{B}})$, which is decreasing as the total budget $\mathbb{B}$ increases. In addition, the regret upper bound is decreased if $\sigma_{\max}^2$ becomes smaller, which implies that the performance of our algorithm is improved if the overall noise level is reduced. Therefore, Theorem 3.1 formalizes the impacts of the experimental properties (i.e., the total budget and the overall noise level) on the performance of BTS-RED-Known.

**Homoscedastic Noise.** In the special case of homoscedastic noise, i.e., $\sigma^2(\boldsymbol{x}) = \sigma_{\text{const}}^2, \forall \boldsymbol{x} \in \mathcal{X}$, then $n_t = \lceil \sigma_{\text{const}}^2/R^2 \rceil \triangleq n_{\text{const}}$ and $b_t = \lfloor \mathbb{B}/n_{\text{const}} \rfloor \triangleq b_0, \forall t \in [T]$. That is, our algorithm reduces

to standard (synchronous) batch TS proposed in [28] where the batch size is $b_0$ and every query is replicated $n_{\text{const}}$ times. In this case, the regret upper bound becomes: $R_T = \widetilde{\mathcal{O}}(R\frac{1}{\sqrt{b_0}}\sqrt{T\Gamma_{Tb_0}}(1 + \sqrt{\Gamma_{Tb_0}}))$ (Appendix B.1).

**Theoretical Guideline on the Choice of $R^2$.** Theorem 3.1 also provides an interesting insight on the choice of the effective noise variance $R^2$. In particular, our regret upper bound depends on $R^2$ through the term $\sqrt{R^2/[\mathbb{B}/(\sigma_{\max}^2/R^2 + 1) - 1]}$.[1] By taking the derivative of this term w.r.t. $R^2$, we have shown (Appendix C) that the value of $R^2$ that minimizes this term is obtained at $R^2 = \sigma_{\max}^2(\sqrt{\mathbb{B}} + 1)/(\mathbb{B} - 1)$. In other words, $R^2$ should be chosen as a fraction of $\sigma_{\max}^2$ (assuming $\mathbb{B} > 4$ s.t. $(\sqrt{\mathbb{B}} + 1)/(\mathbb{B} - 1) < 1$). In this case, increasing the total budget $\mathbb{B}$ naturally encourages more replications. Specifically, increasing $\mathbb{B}$ reduces $(\sqrt{\mathbb{B}} + 1)/(\mathbb{B} - 1)$ and hence decreases the value of $R^2$, which consequently encourages the use of larger $n_t$'s (line 6 of Algo. 1) and allows every selected input to be replicated more times. For example, when the total budget is $\mathbb{B} = 16$, $R^2$ should be chosen as $R^2 = \sigma_{\max}^2/3$; when $\mathbb{B} = 100$, then we have $R^2 = \sigma_{\max}^2/9$. We will follow this theory-inspired choice of $R^2$ in our experiments in Sec. 5 (with slight modifications).

**Improvement over Uniform Sample Allocation.** For the naive baseline of uniform sample allocation (i.e., replicating every input a fixed number $n_0 \leq \mathbb{B}$ of times), the resulting effective observation noise would be $(\sigma_{\max}/\sqrt{n_0})$-sub-Gaussian. This would result in a regret upper bound which can be obtained by replacing the term $\sqrt{R^2/(\mathbb{B}/\lceil\frac{\sigma_{\max}^2}{R^2}\rceil - 1)}$ (Theorem 3.1) by $\sigma_{\max}/\sqrt{n_0}$ (for simplicity, we have ignored the non-integer conditions, i.e., the ceiling operators). Also note that with our optimal choice of $R^2$ (the paragraph above), it can be easily verified that the term $\sqrt{R^2/(\mathbb{B}/\lceil\frac{\sigma_{\max}^2}{R^2}\rceil - 1)}$ (Theorem 3.1) can be simplified to $\sigma_{\max}/\sqrt{\mathbb{B}}$. Therefore, given that $n_0 \leq \mathbb{B}$, our regret upper bound (with the scaling of $\sigma_{\max}/\sqrt{\mathbb{B}}$) is guaranteed to be no worse than that of uniform sample allocation (with the scaling of $\sigma_{\max}/\sqrt{n_0}$).

### 3.2 BTS-RED with Unknown Noise Variance Function

Here we consider the more common scenario where the noise variance function $\sigma^2(\cdot)$ is unknown by extending BTS-RED-Known while preserving its theoretical guarantee.

#### 3.2.1 Modeling of Noise Variance Function

We use a separate GP (denoted as $\mathcal{GP}'$) to model the negative noise variance function $g(\cdot) = -\sigma^2(\cdot)$ and use it to build a high-probability upper bound $U_t^{\sigma^2}(\cdot)$ on the noise variance function $\sigma^2(\cdot)$.[2] After this, we can modify the criteria for selecting $n_t^{(b)}$ (i.e., line 6 of Algo. 1) to be $n_t^{(b)} = \lceil U_t^{\sigma^2}(\boldsymbol{x}_t^{(b)})/R^2 \rceil$, which ensures that $U_t^{\sigma^2}(\boldsymbol{x}_t^{(b)})/n_t^{(b)} \leq R^2$. As a result, the condition of $\sigma^2(\boldsymbol{x}_t^{(b)})/n_t^{(b)} \leq R^2$ is still satisfied (with high probability), which implies that *the observed empirical mean at every queried $\boldsymbol{x}_t^{(b)}$ is still $R-$sub-Gaussian* (Sec. 3.1.1) and theoretical guarantee of Theorem 3.1 is preserved. To construct $\mathcal{GP}'$, we use the (negated) unbiased empirical noise variance $\widetilde{y}_t^{(b)}$ as the noisy observation:

$$\widetilde{y}_t^{(b)} = -1/(n_t^{(b)} - 1) \sum_{n=1}^{n_t^{(b)}} (y_{t,n}^{(b)} - y_t^{(b)})^2 = g(\boldsymbol{x}_t^{(b)}) + \epsilon' \tag{1}$$

where $g(\boldsymbol{x}_t^{(b)}) = -\sigma^2(\boldsymbol{x}_t^{(b)})$ is the negative noise variance at $\boldsymbol{x}_t^{(b)}$, and $\epsilon'$ is the noise. In BTS-RED-Unknown, we use pairs of $\{(\boldsymbol{x}_t^{(b)}, \widetilde{y}_t^{(b)})\}$ to update the posterior of $\mathcal{GP}'$. We impose a minimum number of replications $n_{\min} \geq 2$ for every queried input to ensure reliable estimations of $\widetilde{y}_t^{(b)}$.

#### 3.2.2 Upper Bound on Noise Variance Function

**Assumptions.** Similar to Theorem 3.1, we assume that $g$ lies in an RKHS associated with an SE kernel $k'$: $\|g\|_{\mathcal{H}_{k'}} \leq B'$ for some $B' > 0$, which intuitively assumes that *the (negative) noise variance*

---

[1]To simplify the derivations, we have replaced the term $\lceil \sigma_{\max}^2/R^2 \rceil$ by its upper bound $\sigma_{\max}^2/R^2 + 1$, after which the resulting regret upper bound is still valid.

[2]Here we have modeled $-\sigma^2(\cdot)$ (instead of $\log \sigma^2(\cdot)$ as done by some previous works) because it allows us to naturally derive our theoretical guarantees, and as we show in our experiments (Sec. 5), it indeed allows our algorithms to achieve compelling empirical performances. We will explore modelling $\log \sigma^2(\cdot)$ in future work to see if it leads to further empirical performance gains.

*varies smoothly across the domain $\mathcal{X}$.* We also assume that the noise $\epsilon'$ is $R'$-sub-Gaussian and justify this below by showing that $\epsilon'$ is bounded (with high probability).

$\epsilon'$ **is $R'$-sub-Gaussian.** Since the empirical variance of a Gaussian distribution (1) follows a Chi-squared distribution, we can use the concentration of Chi-squared distributions to show that with probability of $\geq 1 - \alpha$, $\epsilon'$ is bounded within $[L_\alpha, U_\alpha]$, where $L_\alpha = \sigma_{\min}^2(\chi_{n_{\min}-1,\alpha/2}^2/(n_{\min} - 1) - 1), U_\alpha = \sigma_{\max}^2(\chi_{n_{\min}-1,1-\alpha/2}^2/(n_{\min} - 1) - 1)$. Here $\chi_{n_{\min}-1,\eta}^2$ denotes $\eta^{\text{th}}$-quantile of the Chi-squared distribution with $n_{\min} - 1$ degrees of freedom ($\eta = \alpha/2$ or $1 - \alpha/2$). By choosing $\alpha = \delta/(4T\mathbb{B})$ ($\delta$ is from Theorem 3.1), we can ensure that with probability of $\geq 1 - \delta/4$, $\epsilon'$ is bounded within $[L_\alpha, U_\alpha]$ for all $\boldsymbol{x}_t^{[b]}$. In other words, with probability of $\geq 1 - \delta/4$, the noise $\epsilon'$ in (1) is zero-mean and bounded within $[L_\alpha, U_\alpha]$, which indicates that $\epsilon'$ is $R'$-sub-Gaussian with $R' = (U_\alpha - L_\alpha)/2$. More details are given in Appendix E. Note that the value of $R'$ derived here is expected to be overly pessimistic, so, we expect smaller values of $R'$ to be applicable in practice.

**Upper Bound Construction.** With the assumptions of $\|g\|_{\mathcal{H}_{k'}} \leq B'$ and $\epsilon'$ is $R'$-sub-Gaussian, we can construct the upper bound $U_t^{\sigma^2}(\cdot)$. Denote by $\Gamma'_{\tau_{t-1}}$ the maximum information gain about $g$ from any $\tau_{t-1} = \sum_{t'=1}^{t-1} b_{t'}$ observations, define $\beta'_t \triangleq B' + R'\sqrt{2(\Gamma'_{\tau_{t-1}} + 1 + \log(4/\delta))}$, and represent the GP posterior mean and standard deviation for $\mathcal{GP}'$ as $\mu'_{t-1}(\cdot)$ and $\sigma'_{t-1}(\cdot)$. Then we have that

$$|\mu'_{t-1}(\boldsymbol{x}) - g(\boldsymbol{x})| \leq \beta'_t \sigma'_{t-1}(\boldsymbol{x}), \quad \forall \boldsymbol{x} \in \mathcal{X}, t \in [T] \tag{2}$$

with probability of $\geq 1 - \delta/2$. The error probabilities come from applying Theorem 2 of [8] ($\delta/4$) and assuming that $\epsilon'$ is $R'$-sub-Gaussian ($\delta/4$). This implies that $-\sigma^2(\boldsymbol{x}) = g(\boldsymbol{x}) \geq \mu'_{t-1}(\boldsymbol{x}) - \beta'_t \sigma'_{t-1}(\boldsymbol{x})$, and hence $\sigma^2(\boldsymbol{x}) \leq -\mu'_{t-1}(\boldsymbol{x}) + \beta'_t \sigma'_{t-1}(\boldsymbol{x}), \forall \boldsymbol{x} \in \mathcal{X}, t \in [T]$. Therefore, we can choose the upper bound on the noise variance (Sec. 3.2.1) as $U_t^{\sigma^2}(\boldsymbol{x}) = -\mu'_{t-1}(\boldsymbol{x}) + \beta'_t \sigma'_{t-1}(\boldsymbol{x})$.

**BTS-RED-Unknown Algorithm.** To summarize, we can obtain BTS-RED-Unknown (Algo. 3, Appendix F) by modifying the selection criterion of $n_t$ (line 6 of Algo. 1) to be $n_t^{(b)} = \lceil(-\mu'_{t-1}(\boldsymbol{x}_t^{(b)}) + \beta'_t \sigma'_{t-1}(\boldsymbol{x}_t^{(b)}))/R^2\rceil$. As a result, BTS-RED-Unknown enjoys the same regret upper bound as Theorem 3.1 (after replacing $\delta$ in Theorem 3.1 by $\delta/2$). Intuitively, using an upper bound $U_t^{\sigma^2}(\boldsymbol{x}_t^{(b)})$ in the selection of $n_t$ implies that if we are uncertain about the noise variance at some input location $\boldsymbol{x}_t^{(b)}$ (i.e., if $\sigma'_{t-1}(\boldsymbol{x}_t^{(b)})$ is large), we choose to be *conservative* and use a large number of replications $n_t$.

## 4 Mean-Var-BTS-RED

We extend BTS-RED-Unknown (Sec. 3.2) to maximize the mean-variance objective function: $h_\omega(\boldsymbol{x}) = \omega f(\boldsymbol{x}) - (1 - \omega)\sigma^2(\boldsymbol{x})$, to introduce Mean-Var-BTS-RED (Algo. 2). In contrast to BTS-RED-Unknown, Mean-Var-BTS-RED chooses every input query $\boldsymbol{x}_t^{(b)}$ by maximizing the weighted combination of two functions sampled from, respectively, the posteriors of $\mathcal{GP}$ and $\mathcal{GP}'$ (lines 5-6 of Algo. 2), while $n_t$ is chosen (line 7 of Algo. 2) in the same way as BTS-RED-Unknown. This naturally induces a preference for inputs with both large values of $f$ and small values of $\sigma^2(\cdot)$, and hence allows us to derive an upper bound on $R_T^{\text{MV}}$ (proof in Appendix G):

**Theorem 4.1** (Mean-Var-BTS-RED). *With probability of at least $1 - \delta$,*

$$R_T^{MV} = \widetilde{\mathcal{O}}\Big(\frac{e^C \sqrt{T}}{\sqrt{\mathbb{B}/\lceil\frac{\sigma_{\max}^2}{R^2}\rceil - 1}}\Big[\omega R\sqrt{\Gamma_{T\mathbb{B}}}(\sqrt{\Gamma_{T\mathbb{B}}} + \sqrt{C}) + (1 - \omega)R'\sqrt{\Gamma'_{T\mathbb{B}}}(\sqrt{\Gamma'_{T\mathbb{B}}} + \sqrt{C})\Big]\Big).$$

*$C$ is a constant s.t. $\max_{A \subset \mathcal{X}, |A| \leq \mathbb{B}} \mathbb{I}(f; \boldsymbol{y}_A | \boldsymbol{y}_{1:t-1}) \leq C$, $\max_{A \subset \mathcal{X}, |A| \leq \mathbb{B}} \mathbb{I}(g; \widetilde{\boldsymbol{y}}_A | \widetilde{\boldsymbol{y}}_{1:t-1}) \leq C$.*

Note that $\Gamma_{T\mathbb{B}}$ and $\Gamma'_{T\mathbb{B}}$ may differ since the SE kernels $k$ and $k'$, which are used to model $f$ and $g$ respectively, may be different. Similar to Theorem 3.1, if we run uncertainty sampling for a finite number (independent of $T$) of initial iterations using either $k$ or $k'$ (depending on whose lengthscale is smaller), then $C$ can be chosen to be a constant independent of $\mathbb{B}$ and $T$. Refer to Lemma G.6 (Appendix G) for more details. As a result, the regret upper bound in Theorem 4.1 is also sub-linear since both $k$ and $k'$ are SE kernels and hence $\Gamma_{T\mathbb{B}} = \mathcal{O}(\log^{d+1}(T))$ and $\Gamma'_{T\mathbb{B}} = \mathcal{O}(\log^{d+1}(T))$. The regret upper bound can be viewed as a weighted combination of the regrets associated with $f$ and $g$. Intuitively, if $\omega$ is larger (i.e., if we place more emphasis on maximizing $f$ than $g$), then a larger proportion of the regrets is incurred due to our attempt to maximize the function $f$.

**Algorithm 2** Mean-Var-BTS-RED.
1: **for** $t = 1, 2, \ldots, T$ **do**
2:  $b = 0, n_t^{(0)} = 0$
3:  **while** $\sum_{b'=0}^{b} n_t^{(b')} < \mathbb{B}$ **do**
4:   $b \leftarrow b + 1$
5:   Sample $f_t^{(b)}$ from $\mathcal{GP}(\mu_{t-1}(\cdot), \beta_t^2 \sigma_{t-1}^2(\cdot, \cdot))$, and $g_t^{(b)}$ from $\mathcal{GP}'(\mu'_{t-1}(\cdot), {\beta'_t}^2 {\sigma'_{t-1}}^2(\cdot, \cdot))$
6:   $\boldsymbol{x}_t^{(b)} = \arg\max_{\boldsymbol{x} \in \mathcal{X}}[\omega f_t^{(b)}(\boldsymbol{x}) + (1-\omega)g_t^{(b)}(\boldsymbol{x})]$
7:   $n_t^{(b)} = \lceil (-\mu'_{t-1}(\boldsymbol{x}_t^{(b)}) + \beta'_t \sigma'_{t-1}(\boldsymbol{x}_t^{(b)}))/R^2 \rceil$
8:  $b_t = b - 1$
9:  **for** $b \in [b_t]$, query $\boldsymbol{x}_t^{(b)}$ with $n_t^{(b)}$ parallel processes
10:  **for** $b \in [b_t]$, observe $\{y_{t,n}^{(b)}\}_{n \in [n_t^{(b)}]}$. Calculate their mean $y_t^{(b)}$ and (negated) variance $\widetilde{y}_t^{(b)}$ (1)
11:  Use $\{(\boldsymbol{x}_t^{(b)}, y_t^{(b)})\}_{b \in [b_t]}$ to update posterior $\mathcal{GP}$, $\{(\boldsymbol{x}_t^{(b)}, \widetilde{y}_t^{(b)})\}_{b \in [b_t]}$ to update posterior $\mathcal{GP}'$

## 5 Experiments

For BTS-RED-Known and BTS-RED-Unknown which only aim to maximize the objective function $f$, we set $n_{\max} = \mathbb{B}/2$ in the first $T/2$ iterations and $n_{\max} = \mathbb{B}$ subsequently (see Sec. 3.1.1 for more details), and set $n_{\max} = \mathbb{B}$ in all iterations for Mean-Var-BTS-RED. We set $n_{\min} = 2$ unless specified otherwise, however, it is recommended to make $n_{\min}$ larger in experiments where the overall noise variance is large (e.g., we let $n_{\min} = 5$ in Sec. 5.2). We use random search to select the initial inputs instead of the uncertainty sampling initialization method indicated by our theoretical results (Sec. 3.1.2) because previous work [28] and our empirical results show that they lead to similar performances (Fig. 8 in App. H.1). We choose the effective noise variance $R^2$ by following our theoretical guideline in Sec. 3.1.2, i.e., $R^2 = \sigma_{\max}^2(\sqrt{\mathbb{B}}+1)/(\mathbb{B}-1)$ which minimizes the regret upper bound in Theorem 3.1.[3] However, in practice, this choice may not be optimal because we derived it by minimizing an upper bound which is potentially loose (e.g., we have ignored all log factors). So, we introduce a tunable parameter $\kappa > 0$ and choose $R^2$ as $R^2 = \kappa \sigma_{\max}^2(\sqrt{\mathbb{B}} + 1)/(\mathbb{B} - 1)$. As a result, we both enjoy the flexibility of tuning our preference for the overall number of replications (i.e., a smaller $\kappa$ leads to larger $n_t$'s in general) and preserve the ability to automatically adapt to the total budget (via $\mathbb{B}$) and the overall noise level (via $\sigma_{\max}^2$). When the noise variance is unknown (i.e., $\sigma_{\max}^2$ is unknown), we approximate $\sigma_{\max}^2$ by the maximum observed empirical noise variance and update our approximation after every iteration. To demonstrate the robustness of our methods, we only use two values of $\kappa = 0.2$ and $\kappa = 0.3$ in all experiments. Of note, our methods with $\kappa = 0.3$ perform the best in almost all experiments (i.e., green curves in all figures), and $\kappa = 0.2$ also consistently performs well.

Following the common practice of BO [11, 20, 28, 32, 35], we plot the (batch) simple regret or the best observed function value up to an iteration. In all experiments, we compare with the most natural baseline of batch TS with a fixed number of replications. For mean optimization problems (i.e., maximize $f$), we also compare with standard sequential BO algorithms such as GP-UCB and GP-TS, but they are significantly outperformed by both our algorithms and batch TS which are able to exploit batch evaluations (Secs. 5.1 and 5.2). Therefore, we do not expect existing *sequential* algorithms to achieve comparable performances to our algorithms due to their inability to exploit batch evaluations. For mean-variance optimization, we additionally compare with the recently introduced Risk-Averse Heteroscedastic BO (RAHBO) [33] (Sec. 6), which is the state-of-the-art method for risk-averse BO with replications. Some experimental details are postponed to Appendix H.

### 5.1 Synthetic Experiments

We sample two functions from two different GPs with the SE kernel (defined on a discrete 1-D domain within $[0, 1]$) and use them as $f(\cdot)$ and $\sigma^2(\cdot)$, respectively. We use $\mathbb{B} = 50$.

**Mean Optimization.** The mean and noise variance functions used here are visualized in Fig. 1a. This synthetic experiment is used to simulate real-world scenarios where practitioners are *risk-neutral* and hence only aim to select an input with a large mean function value. After every iteration (batch) $t$, an

---

[3]For simplicity, we also follow this guideline from Sec. 3.1.2 to choose $R^2$ for Mean-Var-BTS-RED.

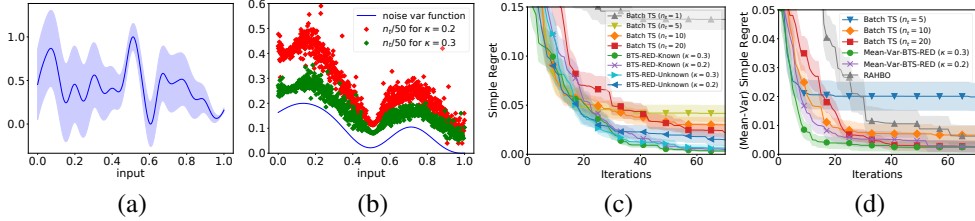

|        |        |        |        |
|:------:|:------:|:------:|:------:|
| (a)    | (b)    | (c)    | (d)    |

Figure 1: (a) Synthetic function for mean optimization (Sec. 5.1). (b) Average number of replications $n_t$ for BTS-RED-Unknown. Results for (c) mean and (d) mean-variance optimization.

algorithm reports the selected input with the largest empirical mean from its observation history, and we evaluate the *simple regret* at iteration $t$ as the difference between the objective function values at the global maximum $\boldsymbol{x}^*$ and at the reported input. To demonstrate the consistency of our performance, we also tested an alternative reporting criteria which reports the input with the larger LCB value in every iteration, and the results (Fig. 7b in Appendix H.1) are consistent with our main results (Fig. 1c). Fig. 1b plots the average $n_t$ (vertical axis) chosen by BTS-RED-Unknown for every queried input (horizontal axis), which shows that larger $n_t$'s are selected for inputs with larger noise variances in general and that a smaller $\kappa = 0.2$ indeed increases our preference for larger $n_t$'s.

The results (simple regrets) are shown in Fig. 1c. As can be seen from the figure, for Batch TS with a fixed $n_t$, smaller values of $n_t$ such as $n_t = 5$ usually lead to faster convergence initially due to the ability to quickly explore more unique inputs, however, their performances deteriorate significantly in the long run due to inaccurate estimations; in contrast, larger $n_t$'s such as $n_t = 20$ result in slower convergence initially yet lead to better performances (than small fixed $n_t$'s) in later stages. Of note, Batch TS with $n_t = 1$ (gray curve) represents standard batch TS ($\mathbb{B} = 50$) without replications [28], which underperforms significantly and hence highlights the importance of replications in experiments with large noise variance. Moreover, our BTS-RED-Known and BTS-RED-Unknown (especially with $\kappa = 0.3$) consistently outperform Batch TS with fixed $n_t$. We also demonstrate our robustness against $\kappa$ in this experiment by showing that our performances are consistent for a wide range of $\kappa$'s (Fig. 7a in App. H.1). In addition, we show that sequential BO algorithms (i.e., GP-TS, GP-UCB, and GP-UCB with heteroscedastic GP) which cannot exploit batch evaluations fail to achieve comparable performances to batch TS, BTS-RED and BTS-RED-Unknown (Fig. 5 in App. H.1).

**Mean-variance Optimization.** Here we evaluate our Mean-Var-BTS-RED. We simulate this scenario with the synthetic function in Fig. 6a (App. H.1), for which the global maximums of the mean and mean-variance ($\omega = 0.3$) objective functions are different (Fig. 6b). After every iteration (batch) $t$, we report the selected input with the largest empirical mean-variance value (i.e., weighted combination of the empirical mean and variance), and evaluate the *mean-variance simple regret* at iteration $t$ as the difference between the values of the mean-variance objective function $h_\omega$ at the the global maximum $\boldsymbol{x}^*_\omega$ and at the reported input. The results (Fig. 1d) show that our Mean-Var-BTS-RED (again especially with $\kappa = 0.3$) outperforms other baselines. Since RAHBO is sequential and uses a fixed number of replications, we use $\mathbb{B} = 50$ replications for every query for a fair comparison. RAHBO underperforms here which is likely due to its inability to leverage batch evaluations.

## 5.2 Real-world Experiments on Precision Agriculture

Plant biologists often need to optimize the growing conditions of plants (e.g., the amount of different nutrients) to increase their yield. The common practice of manually tuning one nutrient at a time is considerably inefficient and hence calls for the use of the sample-efficient method of BO. Unfortunately, plant growths are usually (a) time-consuming and (b) associated with large and heteroscedastic noise. So, *according to plant biologists, in real lab experiments,* (a) *multiple growing conditions are usually tested in parallel* and (b) *every condition is replicated multiple times to get a reliable outcome* [31]. This naturally induces a trade-off between evaluating more unique growing conditions vs. replicating every condition more times, and is hence an ideal application for our algorithms. We tune the pH value (in $[2.5, 6.5]$) and ammonium concentration (denoted as NH3, in $[0, 30000]$ uM). in order to *maximize the leaf area and minimize the tipburn area* after harvest. We perform *real lab experiments* using the input conditions from a regular grid within the 2-D domain, and then use the collected data to learn two separate heteroscedastic GPs for, respectively, leaf area and tipburn area. Each learned GP can output the predicted mean and variance (for leaf area or tipburn area) at every input in the 2-D domain, and can hence be used as the *groundtruth* mean $f(\cdot)$ and noise variance $\sigma^2(\cdot)$ functions. We perform two sets of experiments, with the goal of maximizing (a) the

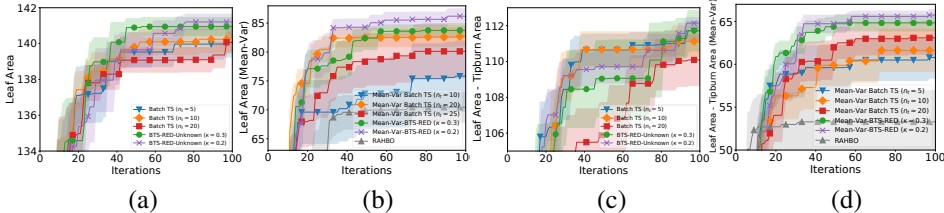

Figure 2: (a) Mean and (b) mean-variance optimization for the leaf area. (c) Mean and (d) mean-variance optimization for the weighted combination of leaf area and negative tipburn area.

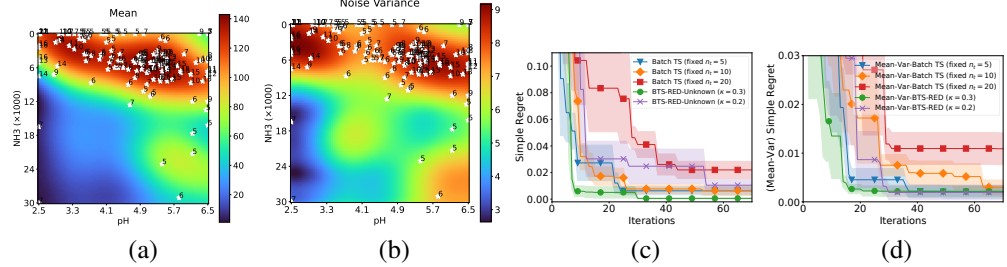

Figure 3: (a) Mean and (b) noise variance functions for leaf area, with some selected queries (stars) and their $n_t$'s. (c) Mean and (d) mean-variance optimization for hyper. tuning of SVM (Sec. 5.3).

leaf area and (b) a weighted combination of the leaf area ($\times 0.8$) and negative tipburn area ($\times 0.2$). For both experiments, we run BTS-RED-Unknown and Mean-Var-BTS-RED to maximize the mean and mean-variance objectives ($\omega = 0.975$), respectively. We set $\mathbb{B} = 50$, $n_{\min} = 5$ and $n_{\max} = 50$.

Fig. 2 shows the results for maximizing the leaf area (a,b) and weighted combination of leaf area and negative tipburn area (c,d). Our BTS-RED-Unknown and Mean-Var-BTS-RED with $\kappa = 0.2$ and $\kappa = 0.3$ consistently outperform Batch TS, as well as RAHBO in Figs. 2b and d. For mean optimization, we also compare with sequential BO methods (Fig. 10 in Appendix H.2), which again are unable to perform comparably with other algorithms that exploit batch evaluations. Figs. 3a and b visualize the groundtruth mean and noise variance functions for the leaf area, including the locations of some queried inputs (the selected inputs after every 4 iterations) and their corresponding $n_t$'s. Similarly, Figs. 9a and b (Appendix H.2) show the queried inputs and the $n_t$'s of Mean-Var-BTS-RED ($\omega = 0.975$), illustrated on heat maps of the mean-variance objective (a) and noise variance functions (b). These figures demonstrate that most of our input queries fall into regions with large (either mean or mean-variance) objective function values (Figs. 3a and 9a) and that $n_t$ is in general larger at those input locations with larger noise variance (Figs. 3b and 9b). We have included GIF animations for Figs. 3 and 9 in the supplementary material. Our results here showcase the capability of our algorithms to improve the efficiency of real-world experimental design problems.

### 5.3 Real-World Experiments on AutoML

Reproducibility is an important desiderata in AutoML problems such as hyperparameter tuning [25], because the performance of a hyperparameter configuration may vary due to a number of factors such as different datasets, parameter initializations, etc. For example, some practitioners may prefer hyperparameter configurations that consistently produce well-performing ML models for different datasets. We adopt the EMNIST dataset which is widely used in multi-task learning [10, 15]. EMNIST consists of images of hand-written characters from different individuals, and each individual corresponds to a separate image classification *task*. Here we tune two SVM hyperparameters: the penalty and RBF kernel parameters, both within $[0.0001, 2]$. We firstly construct a uniform 2-D grid of the two hyeprparameters and then evaluate every input on the grid using 100 tasks (i.e., image classification for 100 different individuals) to record the observed mean and variance as the groundtruth mean and variance. Refer to Figs. 11a, b and c (Appendix H.3) for the constructed mean, variance and mean-variance ($\omega = 0.2$) functions. Fig. 3c and d plot the results ($\mathbb{B} = 50$) for mean (c) and mean-variance (d) optimization. Our BTS-RED-Unknown and Mean-Var-BTS-RED with both $\kappa = 0.2$ and $0.3$ perform competitively (again especially $\kappa = 0.3$), which shows their potential to improve the efficiency and reproducibility of AutoML. RAHBO underperforms significantly (hence omitted from Fig. 3d), which is likely due to the small noise variance (Fig. 11b) which favors methods with small $n_t$'s. Specifically, methods with small $n_t$'s can obtain reliable estimations (due to small

noise variance) while enjoying the advantage of evaluating a large number $b_t$ of unique inputs in every iteration. This makes RAHBO unfavorable since it is a sequential algorithm with $b_t = 1$.

**Experiments Using Different Budgets $\mathbb{B}$.** Here we test the performances of our algorithms with different budgets (i.e., different from the $\mathbb{B} = 50$ used in the main experiments above) using the AutoML experiment. The results (Fig. 12 in App. H.3) show that the performance advantages of our algorithms (again especially with $\kappa = 0.3$) are still consistent with a larger or smaller budget.

**Additional Experiments with Higher-Dimensional Inputs.** To further verify the practicality of our proposed algorithms, here we adopt two additional experiments with higher-dimensional continuous input domains. Specifically, we tune $d = 12$ and $d = 14$ parameters of a controller for a Lunar-Lander task and a robot pushing task, respectively, and both experiments have widely used by previous works on high-dimensional BO [16, 20] (more details in App. H.4). In both experiments, the heteroscedastic noises arise from random environemntal factors. The results (Fig. 13 in App. H.4) show that our algorithms, again especially with $\kappa = 0.3$, still consistently achieve compelling performances.

## 6 Related Works

BO has been extended to the batch setting in recent years [9, 11, 19, 22, 38, 44, 47]. The work of [28] proposed a simple batch TS method by exploiting the inherent randomness of TS. Interestingly, as we discussed in Sec. 3.1.2, the method from [28] is equivalent to a reduced version of our BTS-RED-Known with homoscedastic noise, and our Theorem 3.1 provides a theoretical guarantee on its frequentist regret (in contrast to the Bayesian regret analyzed in [28]). The work of [2] aimed to adaptively choose whether to explore a new query or to replicate a previous query. However, their method requires additional heuristic techniques to achieve replications and hence has no theoretical guarantees, in stark contrast to our simple and principled way for replication selection (Sec. 3). Moreover, their method does not support batch evaluations, and is unable to tackle risk-averse optimization. Recently, [43] proposed to select a batch of queries while balancing exploring new queries and replicating existing ones. However, unlike our simple and principled algorithms, their method requires complicated heuristic procedures for query/replication selection and batch construction, and hence does not have theoretical guarantees. Moreover, their method also only focuses on standard mean optimization and cannot be easily extended for risk-averse optimization.

The work of [23] used a heteroscedastic GP [29] as the surrogate model for risk-averse optimization. The works of [6, 26, 36, 37, 42] considered risk-averse BO, however, these works require the ability to observe and select an environmental variable, which is usually either not explicitly defined or uncontrollable in practice (e.g., our experiments in Sec. 5). The recent work of [33] modified BO to maximize the mean-variance objective and derived theoretical guarantees using results from [30]. Their method uses a heteroscedastic GP as the surrogate model and employs another (homoscedastic) GP to model the observation noise variance, in which the second GP is learned by replicating every query for a fixed predetermined number of times. Importantly, all of these works on risk-averse BO have focused only on the sequential setting without support for batch evaluations. Replicating the selected inputs in BO multiple times has also been adopted by the recent works of [7, 13], which have shown that replication can lead to comparable or better theoretical and empirical performances of BO.

## 7 Conclusion

We have introduced the BTS-RED framework, which can trade-off between evaluating more unique conditions vs. replicating each condition more times and can perform risk-averse optimization. We derive theoretical guarantees for our methods to show that they are no-regret, and verify their empirical effectiveness in real-world precision agriculture and AutoML experiments. A potential limitation is that we use a heuristic (rather than principled) technique to handle unused budgets in an iteration (last paragraph of Sec. 3.1.1). Another interesting future work is to incorporate our technique of using an adaptive number of replications (depending on the noise variance) into other batch BO algorithms [18, 22] to further improve their performances. Moreover, it is also interesting to combine our method with the recent line of work on neural bandits [16, 17], which may expand the application of our method to more AI4Science problems.

## Acknowledgements and Disclosure of Funding

This research/project is supported by A*STAR under its RIE2020 Advanced Manufacturing and Engineering (AME) Programmatic Funds (Award A20H6b0151) and its RIE2020 Advanced Manufacturing and Engineering (AME) Industry Alignment Fund – Pre Positioning (IAF-PP) (Award A19E4a0101).

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

## A Expressions of GP Posterior

Following the notations in the main text (Sec. 2), we index every queried input $\boldsymbol{x}_t^{[b]}$ and observed output $y_t^{[b]}$ via an iteration index $t$ and a batch index $b$. Define $\mathcal{I}_{t-1}$ the collection of indices of the queried inputs in the first $t-1$ iterations: $\mathcal{I}_{t-1} = \{(t',b)\}_{t'\in[t-1],b\in[b_{t'}]}$. Note that according to our notations in the main text, the cardinality of $\mathcal{I}_{t-1}$ is $|\mathcal{I}_{t-1}| = \tau_{t-1} = \sum_{t'=1}^{t-1} b_{t'}$. Then, the GP posterior for the objective function $f$ in iteration $t$ can be represented as $\mathcal{GP}(\mu_{t-1}(\cdot), \sigma_{t-1}^2(\cdot,\cdot))$, where

$$\begin{aligned}
\mu_{t-1}(\boldsymbol{x}) &\triangleq \boldsymbol{k}_{t-1}(\boldsymbol{x})^\top (\boldsymbol{K}_{t-1} + \lambda\mathbf{I})^{-1}\boldsymbol{y}_{t-1}, \\
\sigma_{t-1}^2(\boldsymbol{x}, \boldsymbol{x}') &\triangleq k(\boldsymbol{x}, \boldsymbol{x}') - \boldsymbol{k}_{t-1}(\boldsymbol{x})^\top (\boldsymbol{K}_{t-1} + \lambda\mathbf{I})^{-1}\boldsymbol{k}_{t-1}(\boldsymbol{x}'),
\end{aligned} \tag{3}$$

in which $\boldsymbol{k}_{t-1}(\boldsymbol{x}) \triangleq [k(\boldsymbol{x}, \boldsymbol{x}_{t'}^{[b]})]_{(t',b)\in\mathcal{I}_{t-1}}^\top$. $\boldsymbol{y}_{t-1} \triangleq (y_{t'}^{[b]})_{(t',b)\in\mathcal{I}_{t-1}}^\top$ in which $y_{t'}^{[b]} = (1/n_{t'}^{[b]})\sum_{n=1}^{n_{t'}^{[b]}} y_{t',n}^{[b]}$ represents the empirical mean at the input $\boldsymbol{x}_{t'}^{[b]}$ calculated using the $n_{t'}^{[b]}$ replications. $\mathbf{K}_{t-1} \triangleq (k(\mathbf{x}_{t'}^{[b]}, \mathbf{x}_{t''}^{[b']}))_{(t',b)\in\mathcal{I}_{t-1},(t'',b')\in\mathcal{I}_{t-1}}$ and $\lambda > 0$ is a regularization parameter and will need to be set to $\lambda = 1 + 2/T$ in order for our theoretical results to hold [8].

## B Proof of Theorem 3.1

Denote by $\tau_{t-1}$ the total number of observations (input-output pairs) up to and including iteration $t-1$: $\tau_{t-1} \triangleq \sum_{t'=1}^{t-1} b_{t'}$. This immediately implies that $t-1 \leq \tau_{t-1} \leq \mathbb{B}(t-1)$. We use $\mathcal{F}_{t-1}$ to denote the history of all $\tau_{t-1}$ observations up to iteration $t-1$. Denote by $b_t$ the batch size in iteration $t$. Note that conditioned on $\mathcal{F}_{t-1}$, $b_t$ is a random variable, which is in contrast with standard batch BO in which the batch size is usually fixed. Here we use $\mu_{t-1}(\boldsymbol{x})$ and $\sigma_{t-1}(\boldsymbol{x})$ to denote the GP posterior mean and standard deviation conditioned on *all* $\tau_{t-1}$ observations up to (and including) iteration $t-1$. Moreover, denote by $\sigma_{t-1,b'}(\boldsymbol{x})$ the GP posterior standard deviation after *additionally* conditioning on the first $b' = 0, \ldots, b_t - 1$ selected inputs in iteration $t$. Note that $\sigma_{t-1,0}(\boldsymbol{x}) = \sigma_{t-1}(\boldsymbol{x})$ according to our definitions. Define $\beta_t \triangleq B + R\sqrt{2(\Gamma_{\tau_{t-1}} + 1 + \log(2/\delta))}$ and $c_t \triangleq \beta_t(1 + \sqrt{2\log(\mathbb{B}|\mathcal{X}|t^2)})$.

**Lemma B.1.** *Let $\delta \in (0,1)$. Define $E^f(t)$ as the event that $|\mu_{t-1}(\boldsymbol{x}) - f(\boldsymbol{x})| \leq \beta_t\sigma_{t-1}(\boldsymbol{x})$ for all $\boldsymbol{x} \in \mathcal{X}$. We have that $\mathbb{P}\left[E^f(t)\right] \geq 1 - \delta/2$ for all $t \geq 1$.*

Lemma B.1 is a consequence of Theorem 2 of the work of [8].

**Lemma B.2.** *Define $E^{f_t}(t)$ as the event: $|f_t^{[b]}(\boldsymbol{x}) - \mu_{t-1}(\boldsymbol{x})| \leq \beta_t\sqrt{2\log(\mathbb{B}|\mathcal{X}|t^2)}\sigma_{t-1}(\boldsymbol{x})$, $\forall\boldsymbol{x} \in \mathcal{X}, \forall b \in [b_t]$. We have that $\mathbb{P}\left[E^{f_t}(t)|\mathcal{F}_{t-1}\right] \geq 1 - 1/t^2$ for any possible filtration $\mathcal{F}_{t-1}$.*

*Proof.* According to Lemma B4 of [24], in iteration $t$, for a particular $b$ and $\boldsymbol{x}$, we have that

$$|f_t^{[b]}(\boldsymbol{x}) - \mu_{t-1}(\boldsymbol{x})| \leq \beta_t\sqrt{2\log(1/\delta)}\sigma_{t-1}(\boldsymbol{x}), \tag{4}$$

with probability of $\geq 1 - \delta$. Replacing $\delta$ by $\delta/(\mathbb{B}|\mathcal{X}|)$ and taking a union bound over all $\boldsymbol{x} \in \mathcal{X}$ and all $b \in [b_t]$ gives us:

$$|f_t^{[b]}(\boldsymbol{x}) - \mu_{t-1}(\boldsymbol{x})| \leq \beta_t\sqrt{2\log(\mathbb{B}|\mathcal{X}|/\delta)}\sigma_{t-1}(\boldsymbol{x}), \qquad \forall\boldsymbol{x} \in \mathcal{X}, b \in [b_t], \tag{5}$$

which holds with probability of $\geq 1 - \frac{\delta}{\mathbb{B}|\mathcal{X}|} \times |\mathcal{X}|b_t \geq 1 - \delta$, because $b_t \leq \mathbb{B}$. Further replacing $\delta$ by $1/t^2$ completes the proof. $\qquad\square$

Next, we define the set of *saturated points*.

**Definition B.3.** Define the set of saturated points at iteration $t$ as

$$S_t = \{\boldsymbol{x} \in \mathcal{X} : \Delta(\boldsymbol{x}) > c_t\sigma_{t-1}(\boldsymbol{x})\},$$

in which $\Delta(\boldsymbol{x}) = f(\boldsymbol{x}^*) - f(\boldsymbol{x})$ and $\boldsymbol{x}^* \in \arg\max_{\boldsymbol{x}\in\mathcal{X}} f(\boldsymbol{x})$.

The next auxiliary lemma will be needed shortly to lower-bound the probability that an unsaturated point is selected.

**Lemma B.4.** *For any filtration $\mathcal{F}_{t-1}$, conditioned on the events $E^f(t)$, we have that $\forall \boldsymbol{x} \in \mathcal{X}, b \in [b_t]$,*

$$\mathbb{P}\left(f_t^{[b]}(\boldsymbol{x}) > f(\boldsymbol{x})|\mathcal{F}_{t-1}\right) \geq p, \tag{6}$$

*in which $p = \frac{1}{4e\sqrt{\pi}}$.*

*Proof.* For any $b \in [b_t]$, we have that

$$
\begin{aligned}
\mathbb{P}\left(f_t^{[b]}(\boldsymbol{x}) > f(\boldsymbol{x})|\mathcal{F}_{t-1}\right) &= \mathbb{P}\left(\frac{f_t^{[b]}(\boldsymbol{x}) - \mu_{t-1}(\boldsymbol{x})}{\beta_t \sigma_{t-1}(\boldsymbol{x})} > \frac{f(\boldsymbol{x}) - \mu_{t-1}(\boldsymbol{x})}{\beta_t \sigma_{t-1}(\boldsymbol{x})}\bigg|\mathcal{F}_{t-1}\right) \\
&\geq \mathbb{P}\left(\frac{f_t^{[b]}(\boldsymbol{x}) - \mu_{t-1}(\boldsymbol{x})}{\beta_t \sigma_{t-1}(\boldsymbol{x})} > \frac{|f(\boldsymbol{x}) - \mu_{t-1}(\boldsymbol{x})|}{\beta_t \sigma_{t-1}(\boldsymbol{x})}\bigg|\mathcal{F}_{t-1}\right) \\
&\geq \mathbb{P}\left(\frac{f_t^{[b]}(\boldsymbol{x}) - \mu_{t-1}(\boldsymbol{x})}{\beta_t \sigma_{t-1}(\boldsymbol{x})} > 1\bigg|\mathcal{F}_{t-1}\right) \\
&\geq \frac{e^{-1}}{4\sqrt{\pi}}.
\end{aligned}
\tag{7}
$$

The second last inequality results from Lemma B.1, and the last inequality follows because $f_t^{[b]}(\boldsymbol{x})$ follows a Gaussian distribution because $f_t^{[b]} \sim \mathcal{GP}(\mu_{t-1}(\cdot), \beta_t^2 \sigma_{t-1}^2(\cdot))$. Lastly, since all $f_t^{[b]}$'s are sampled in the same way: $f_t^{[b]} \sim \mathcal{GP}(\mu_{t-1}(\cdot), \beta_t^2 \sigma_{t-1}^2(\cdot))$, the proof above holds for all $b \in [b_t]$. $\quad\square$

The next lemma shows that the probability that an unsaturated input is selected can be lower-bounded.

**Lemma B.5.** *For any filtration $\mathcal{F}_{t-1}$, conditioned on the event $E^f(t)$, we have that*

$$\mathbb{P}\left(\boldsymbol{x}_t^{[b]} \in \mathcal{X} \setminus S_t, |\mathcal{F}_{t-1}\right) \geq p - 1/t^2, \qquad \forall b \in [b_t].$$

*Proof.* For every $b \in [b_t]$,

$$\mathbb{P}\left(\boldsymbol{x}_t^{[b]} \in \mathcal{X} \setminus S_t|\mathcal{F}_{t-1}\right) \geq \mathbb{P}\left(f_t^{[b]}(\boldsymbol{x}^*) > f_t^{[b]}(\boldsymbol{x}), \forall \boldsymbol{x} \in S_t|\mathcal{F}_{t-1}\right), \tag{8}$$

which holds $\forall b \in [b_t]$. The validity of the inequality above can be seen by noting that $\boldsymbol{x}^*$ is *always unsaturated*, because $\Delta(\boldsymbol{x}^*) = f(\boldsymbol{x}^*) - f(\boldsymbol{x}^*) = 0 < c_t \sigma_{t-1}(\boldsymbol{x})$. As a result, if the event on the right hand side holds (i.e., if $f_t^{[b]}(\boldsymbol{x}^*) > f_t^{[b]}(\boldsymbol{x}), \forall \boldsymbol{x} \in S_t$), then the event on the left hand side is guaranteed to hold because $\boldsymbol{x}_t^{[b]}$ is selected by $\boldsymbol{x}_t^{[b]} = \arg\max_{\boldsymbol{x} \in \mathcal{X}} f_t^{[b]}(\boldsymbol{x})$ which ensures that an unsaturated input will be selected.

Next, we assume that both events $E^f(t)$ and $E^{f_t}(t)$ are true, which allows us to derive an upper bound on $f_t^{[b]}(\boldsymbol{x})$ for all $\boldsymbol{x} \in S_t$ and for all $b \in [b_t]$:

$$f_t^{[b]}(\boldsymbol{x}) \leq f(\boldsymbol{x}) + c_t \sigma_{t-1}(\boldsymbol{x}) \leq f(\boldsymbol{x}) + \Delta(\boldsymbol{x}) = f(\boldsymbol{x}) + f(\boldsymbol{x}^*) - f(\boldsymbol{x}) = f(\boldsymbol{x}^*), \tag{9}$$

where the first inequality follows from Lemma B.1 and Lemma B.2, and the second inequality results from Definition B.3. Therefore, (9) implies that for every $b \in [b_t]$, if both both events $E^f(t)$ and $E^{f_t}(t)$ hold, we have that

$$\mathbb{P}\left(f_t^{[b]}(\boldsymbol{x}^*) > f_t^{[b]}(\boldsymbol{x}), \forall \boldsymbol{x} \in S_t|\mathcal{F}_{t-1}\right) \geq \mathbb{P}\left(f_t^{[b]}(\boldsymbol{x}^*) > f(\boldsymbol{x}^*)|\mathcal{F}_{t-1}\right). \tag{10}$$

Next, conditioning only on the event $E^f(t)$, for every $b \in [b_t]$, we can show that

$$
\begin{aligned}
\mathbb{P}\left(\boldsymbol{x}_t^{[b]} \in \mathcal{X} \setminus S_t|\mathcal{F}_{t-1}\right) &\geq \mathbb{P}\left(f_t^{[b]}(\boldsymbol{x}^*) > f_t^{[b]}(\boldsymbol{x}), \forall \boldsymbol{x} \in S_t|\mathcal{F}_{t-1}\right) \\
&\overset{(a)}{\geq} \mathbb{P}\left(f_t^{[b]}(\boldsymbol{x}^*) > f(\boldsymbol{x}^*)|\mathcal{F}_{t-1}\right) - \mathbb{P}\left(\overline{E^{f_t}(t)}|\mathcal{F}_{t-1}\right) \\
&\overset{(b)}{\geq} p - 1/t^2,
\end{aligned}
\tag{11}
$$

which holds for all $b \in [b_t]$. $\quad\square$

Next, we use the following Lemma to connect the GP posterior standard deviation given all observations in the first $t-1$ iterations (i.e., $\sigma_{t-1}(\cdot)$) with the conditional information gain from the selected input queries in the $t^{\text{th}}$ iteration (batch).

**Lemma B.6.** *Define $C_2 = \frac{2}{\log(1+\lambda^{-1})}$. Denote all $\tau_{t-1}$ observations from iterations (batches) 1 to $t-1$ as $\boldsymbol{y}_{1:t-1}$, and the $b_t$ observations in the $t^{th}$ batch as $\boldsymbol{y}_t$. Then we have that*

$$\sum_{b=1}^{b_t} \sigma_{t-1}(\boldsymbol{x}_t^{[b]}) \leq e^C \sqrt{C_2 b_t \mathbb{I}(f; \boldsymbol{y}_t | \boldsymbol{y}_{1:t-1})}.$$

*Proof.* Note that as has been described in the main text, the constant $C$ is chosen such that:

$$\max_{A \subset \mathcal{X}, |A| \leq \mathbb{B}} \mathbb{I}\left(f; \boldsymbol{y}_A | \boldsymbol{y}_{1:t-1}\right) \leq C, \forall t \geq 1. \tag{12}$$

Denote by $\boldsymbol{y}_{t,1:b-1}$ the first $b-1$ observations within the $t^{\text{th}}$ batch, then for $b > 1$,

$$
\begin{aligned}
\frac{\sigma_{t-1}(\boldsymbol{x})}{\sigma_{t-1,b-1}(\boldsymbol{x})} &= \exp\left(\mathbb{I}(f(\boldsymbol{x}); \boldsymbol{y}_{t,1:b-1} | \boldsymbol{y}_{1:t-1})\right) \\
&\leq \exp\left(\mathbb{I}(f; \boldsymbol{y}_{t,1:b-1} | \boldsymbol{y}_{1:t-1})\right) \\
&\leq \exp\left(\max_{A \subset \mathcal{X}, |A| \leq b-1} \mathbb{I}(f; \boldsymbol{y}_A | \boldsymbol{y}_{1:t-1})\right) \\
&\leq \exp\left(\max_{A \subset \mathcal{X}, |A| \leq \mathbb{B}} \mathbb{I}(f; \boldsymbol{y}_A | \boldsymbol{y}_{1:t-1})\right) \\
&\leq \exp(C).
\end{aligned}
\tag{13}
$$

Also note that when $b = 1$, $\sigma_{t-1}(\boldsymbol{x})/\sigma_{t-1,b-1}(\boldsymbol{x}) = 1 \leq \exp(C)$. Therefore, we have that

$$
\begin{aligned}
\sum_{b=1}^{b_t} \sigma_{t-1}(\boldsymbol{x}_t^{[b]}) &\leq \sum_{b=1}^{b_t} e^C \sigma_{t-1,b-1}(\boldsymbol{x}_t^{[b]}) \leq e^C \sqrt{b_t \sum_{b=1}^{b_t} \sigma_{t-1,b-1}^2(\boldsymbol{x}_t^{[b]})} \\
&\leq e^C \sqrt{b_t \sum_{b=1}^{b_t} \frac{1}{\log(1+\lambda^{-1})} \log\left(1 + \lambda^{-1}\sigma_{t-1,b-1}^2(\boldsymbol{x}_t^{[b]})\right)} \\
&= e^C \sqrt{C_2 b_t \frac{1}{2} \sum_{b=1}^{b_t} \log\left(1 + \lambda^{-1}\sigma_{t-1,b-1}^2(\boldsymbol{x}_t^{[b]})\right)} \\
&= e^C \sqrt{C_2 b_t \mathbb{I}\left(f; \boldsymbol{y}_t | \boldsymbol{y}_{1:t-1}\right)}.
\end{aligned}
\tag{14}
$$

The second inequality makes use of the Cauchy–Schwarz inequality, and the last equality follows from the definition of information gain. $\qquad\square$

The next Lemma gives an upper bound on the expected batch regret in iteration $t$: $\min_{b \in [b_t]} r_t^{[b]} = \min_{b \in [b_t]} (f(\boldsymbol{x}^*) - f(\boldsymbol{x}_t^{[b]}))$.

**Lemma B.7.** *For any filtration $\mathcal{F}_{t-1}$, conditioned on the event $E^f(t)$, we have that*

$$\mathbb{E}\left[\min_{b \in [b_t]} r_t^{[b]} \Big| \mathcal{F}_{t-1}\right] \leq c_t e^C \left(1 + \frac{2}{p - 1/t^2}\right) \mathbb{E}\left[\sqrt{\frac{1}{b_t} C_2 \mathbb{I}\left(f; \boldsymbol{y}_t | \boldsymbol{y}_{1:t-1}\right)} \Big| \mathcal{F}_{t-1}\right] + \frac{2B}{t^2},$$

*in which $r_t^{[b]} = f(\boldsymbol{x}^*) - f(\boldsymbol{x}_t^{[b]})$.*

*Proof.* To begin with, we define $\overline{\boldsymbol{x}}_t$ as the unsaturated input at iteration $t$ (after the first $t-1$ iterations) with the smallest (posterior) standard deviation:

$$\overline{\boldsymbol{x}}_t \triangleq \arg\min_{\boldsymbol{x}\in\mathcal{X}\setminus S_t} \sigma_{t-1}(\boldsymbol{x}). \tag{15}$$

Following this definition, for any $\mathcal{F}_{t-1}$ such that $E^f(t)$ is true, $\forall b\in[b_t]$, we have that

$$\mathbb{E}\left[\sigma_{t-1}(\boldsymbol{x}_t^{[b]})|\mathcal{F}_{t-1}\right] \geq \mathbb{E}\left[\sigma_{t-1}(\boldsymbol{x}_t^{[b]})|\mathcal{F}_{t-1}, \boldsymbol{x}_t^{[b]}\in\mathcal{X}\setminus S_t\right] \mathbb{P}\left(\boldsymbol{x}_t^{[b]}\in\mathcal{X}\setminus S_t|\mathcal{F}_{t-1}\right) \tag{16}$$
$$\geq \sigma_{t-1}(\overline{\boldsymbol{x}}_t)(p-1/t^2),$$

Now we condition on both events $E^f(t)$ and $E^{f_t}(t)$, and analyze the instantaneous regret as:

$$\min_{b\in[b_t]} r_t^{[b]} \leq \frac{1}{b_t}\sum_{b=1}^{b_t} r_t^{[b]} = \frac{1}{b_t}\sum_{b=1}^{b_t}\Delta(\boldsymbol{x}_t^{[b]}) = \frac{1}{b_t}\sum_{b=1}^{b_t}\left[f(\boldsymbol{x}^*) - f(\overline{\boldsymbol{x}}_t) + f(\overline{\boldsymbol{x}}_t) - f(\boldsymbol{x}_t^{[b]})\right]$$

$$\leq \frac{1}{b_t}\sum_{b=1}^{b_t}\left[\Delta(\overline{\boldsymbol{x}}_t) + f_t^{[b]}(\overline{\boldsymbol{x}}_t) + c_t\sigma_{t-1}(\overline{\boldsymbol{x}}_t) - f_t^{[b]}(\boldsymbol{x}_t^{[b]}) + c_t\sigma_{t-1}(\boldsymbol{x}_t^{[b]})\right] \tag{17}$$

$$\leq \frac{1}{b_t}\sum_{b=1}^{b_t}\left[c_t\sigma_{t-1}(\overline{\boldsymbol{x}}_t) + c_t\sigma_{t-1}(\overline{\boldsymbol{x}}_t) + c_t\sigma_{t-1}(\boldsymbol{x}_t^{[b]}) + f_t^{[b]}(\overline{\boldsymbol{x}}_t) - f_t^{[b]}(\boldsymbol{x}_t^{[b]})\right]$$

$$\leq \frac{1}{b_t}\sum_{b=1}^{b_t}\left[c_t(2\sigma_{t-1}(\overline{\boldsymbol{x}}_t) + \sigma_{t-1}(\boldsymbol{x}_t^{[b]}))\right],$$

in which the second inequality results from Lemma B.1 and Lemma B.2, the third inequality follows since $\overline{\boldsymbol{x}}_t$ is unsaturated, and the last inequality follows from the policy in which $\boldsymbol{x}_t^{[b]}$ is selected, i.e., $\boldsymbol{x}_t^{[b]} = \arg\max_{\boldsymbol{x}\in\mathcal{X}} f_t^{[b]}(\boldsymbol{x})$.

Now we separately consider the two cases where the event $E^{f_t}(t)$ is true and false:

$$\mathbb{E}\left[\min_{b\in[b_t]} r_t^{[b]}\Big|\mathcal{F}_{t-1}\right] \leq \mathbb{E}\left[\frac{1}{b_t}\sum_{b=1}^{b_t}\left[c_t(2\sigma_{t-1}(\overline{\boldsymbol{x}}_t) + \sigma_{t-1}(\boldsymbol{x}_t^{[b]}))\right]\Big|\mathcal{F}_{t-1}\right] + 2B\mathbb{P}\left[\overline{E^{f_t}(t)}|\mathcal{F}_{t-1}\right]$$

$$\leq \mathbb{E}\left[\frac{1}{b_t}\sum_{b=1}^{b_t}\left[\frac{2c_t}{p-1/t^2}\sigma_{t-1}(\boldsymbol{x}_t^{[b]}) + c_t\sigma_{t-1}(\boldsymbol{x}_t^{[b]})\right]\Big|\mathcal{F}_{t-1}\right] + \frac{2B}{t^2}$$

$$\leq c_t\left(1 + \frac{2}{p-1/t^2}\right)\mathbb{E}\left[\frac{1}{b_t}\sum_{b=1}^{b_t}\sigma_{t-1}(\boldsymbol{x}_t^{[b]})\Big|\mathcal{F}_{t-1}\right] + \frac{2B}{t^2}$$

$$\leq c_t e^C\left(1 + \frac{2}{p-1/t^2}\right)\mathbb{E}\left[\frac{1}{b_t}\sqrt{C_2 b_t\mathbb{I}\left(f;\boldsymbol{y}_t|\boldsymbol{y}_{1:t-1}\right)}\Big|\mathcal{F}_{t-1}\right] + \frac{2B}{t^2}$$

$$\leq c_t e^C\left(1 + \frac{10}{p}\right)\mathbb{E}\left[\sqrt{\frac{1}{b_t}C_2\mathbb{I}\left(f;\boldsymbol{y}_t|\boldsymbol{y}_{1:t-1}\right)}\Big|\mathcal{F}_{t-1}\right] + \frac{2B}{t^2}, \tag{18}$$

where the second inequality follows from equation (16), the fourth inequality results from Lemma B.6, and the last inequality follows since $\frac{2}{p-1/t^2} \leq \frac{10}{p}$.

$\square$

**Definition B.8.** Define $Y_0 = 0$, and for all $t = 1,\ldots,T$,

$$\overline{r}_t = \mathbb{I}\{E^f(t)\}\min_{b\in[b_t]} r_t^{[b]},$$

$$X_t = \overline{r}_t - c_t e^C\left(1 + \frac{10}{p}\right)\sqrt{\frac{1}{b_t}C_2\mathbb{I}\left(f;\boldsymbol{y}_t|\boldsymbol{y}_{1:t-1}\right)} - \frac{2B}{t^2}$$

$$Y_t = \sum_{s=1}^{t} X_s.$$

**Lemma B.9.** *Conditioned on Lemma B.7 (i.e., with probability of $\geq 1 - \delta/2$), $(Y_t : t = 0, \dots, T)$ is a super-martingale with respect to the filtration $\mathcal{F}_t$.*

*Proof.*

$$\mathbb{E}[Y_t - Y_{t-1}|\mathcal{F}_{t-1}] = \mathbb{E}[X_t|\mathcal{F}_{t-1}]$$

$$= \mathbb{E}[\bar{r}_t - c_t e^C \left(1 + \frac{10}{p}\right) \sqrt{\frac{1}{b_t} C_2 \mathbb{I}\left(f; \boldsymbol{y}_t|\boldsymbol{y}_{1:t-1}\right)} - \frac{2B}{t^2}|\mathcal{F}_{t-1}]$$

$$= \mathbb{E}[\bar{r}_t|\mathcal{F}_{t-1}] - \mathbb{E}[c_t e^C \left(1 + \frac{10}{p}\right) \sqrt{\frac{1}{b_t} C_2 \mathbb{I}\left(f; \boldsymbol{y}_t|\boldsymbol{y}_{1:t-1}\right)} + \frac{2B}{t^2}|\mathcal{F}_{t-1}] \leq 0. \tag{19}$$

When the event $E^f(t)$ holds, then $\bar{r}_t = \min_{b \in [b_t]} r_t^{[b]}$ and the inequality follows from Lemma B.7; when $E^f(t)$ does not hold, $\bar{r}_t = 0$ and hence the inequality trivially holds.

$\square$

**Lemma B.10.** *Define $C_0 \triangleq \frac{1}{\mathbb{B}/\lceil \frac{\sigma_{\max}^2}{R^2} \rceil - 1}$. Given $\delta \in (0, 1)$, then with probability of at least $1 - \delta$,*

$$R_T \leq e^C \left(1 + \frac{10}{p}\right) c_T \sqrt{C_0 C_2 \Gamma_T T} + \frac{B\pi^2}{3} + \left(4B + c_T e^C \left(1 + \frac{10}{p}\right) \sqrt{CC_0 C_2}\right) \sqrt{2 \log(2/\delta)T}.$$

*Proof.* To begin with, let's derive a lower bound on $b_t$. Firstly, note that $n_t \leq \lceil \frac{\sigma_{\max}^2}{R^2} \rceil$. This implies that $b_t \geq \mathbb{B}/\lceil \frac{\sigma_{\max}^2}{R^2} \rceil - 1$. Next, we need to derive an upper bound on $|Y_t - Y_{t-1}|$ which will be used when we apply the Azuma-Hoeffding's inequality:

$$|Y_t - Y_{t-1}| = |X_t| \leq |\bar{r}_t| + c_t e^C \left(1 + \frac{10}{p}\right) \sqrt{\frac{1}{b_t} C_2 \mathbb{I}\left(f; \boldsymbol{y}_t|\boldsymbol{y}_{1:t-1}\right)} + \frac{2B}{t^2}$$

$$\leq 2B + c_t e^C \left(1 + \frac{10}{p}\right) \sqrt{\frac{CC_2}{b_t}} + 2B \tag{20}$$

$$\leq 4B + c_t e^C \left(1 + \frac{10}{p}\right) \sqrt{\frac{CC_2}{b_t}},$$

where the second inequality follows because

$$\mathbb{I}\left(f; \boldsymbol{y}_t|\boldsymbol{y}_{1:t-1}\right) \leq \max_{A \subset \mathcal{X}, |A| \leq b_t} \mathbb{I}\left(f; \boldsymbol{y}_A|\boldsymbol{y}_{1:t-1}\right) \leq \max_{A \subset \mathcal{X}, |A| \leq \mathbb{B}} \mathbb{I}\left(f; \boldsymbol{y}_A|\boldsymbol{y}_{1:t-1}\right) \leq C, \forall t \geq 1.$$

Next, we will need the following result to connect the sum of condition information gains to the maximum information gain:

$$\sum_{t=1}^{T} \sqrt{\mathbb{I}\left(f; \boldsymbol{y}_t|\boldsymbol{y}_{1:t-1}\right)} \leq \sqrt{T \sum_{t=1}^{T} \mathbb{I}\left(f; \boldsymbol{y}_t|\boldsymbol{y}_{1:t-1}\right)} = \sqrt{T \mathbb{I}\left(f; \boldsymbol{y}_{1:T}\right)} \leq \sqrt{T \Gamma_{T\mathbb{B}}}, \tag{21}$$

in which the first inequality results from the Cauchy–Schwarz inequality, the second inequality follows from the chain rule of conditional information gain, and the last inequality makes use of the

fact that $\tau_T \leq T\mathbb{B}$. Subsequently, we are ready to upper-bound the batch cumulative regret:

$$
\sum_{t=1}^{T} \bar{r}_t \leq \sum_{t=1}^{T} c_t e^C \left(1 + \frac{10}{p}\right) \sqrt{\frac{1}{b_t} C_2 \mathbb{I}\left(f; \boldsymbol{y}_t | \boldsymbol{y}_{1:t-1}\right)} + \sum_{t=1}^{T} \frac{2B}{t^2} +
$$

$$
\sqrt{2\log(2/\delta) \sum_{t=1}^{T} \left(4B + c_t e^C \left(1 + \frac{10}{p}\right) \sqrt{\frac{CC_2}{b_t}}\right)^2}
$$

$$
\leq e^C \left(1 + \frac{10}{p}\right) c_T \sqrt{\frac{C_2}{\mathbb{B}/\lceil \frac{\sigma_{\max}^2}{R^2} \rceil - 1}} \sum_{t=1}^{T} \sqrt{\mathbb{I}\left(f; \boldsymbol{y}_t | \boldsymbol{y}_{1:t-1}\right)} + \frac{B\pi^2}{3} +
$$

$$
\left(4B + c_T e^C \left(1 + \frac{10}{p}\right) \sqrt{\frac{CC_2}{\mathbb{B}/\lceil \frac{\sigma_{\max}^2}{R^2} \rceil - 1}}\right) \sqrt{2\log(2/\delta)T}
$$

$$
\leq e^C \left(1 + \frac{10}{p}\right) c_T \sqrt{C_0 C_2} \sqrt{T\Gamma_{T\mathbb{B}}} + \frac{B\pi^2}{3} + \left(4B + c_T e^C \left(1 + \frac{10}{p}\right) \sqrt{CC_0 C_2}\right) \sqrt{2\log(2/\delta)T}.
$$
(22)

The first inequality results from the Azuma-Hoeffding's inequality, the second inequality makes use of the lower bound on $b_t$: $b_t \geq \mathbb{B}/\lceil \frac{\sigma_{\max}^2}{R^2} \rceil - 1$, and the last inequality follows from equation (21). Equation (22) holds with probability of $\geq 1 - \delta/2$ according to the Azuma-Hoeffding's inequality. Then, we also have that $r_t = \bar{r}_t, \forall t \geq 1$ with probability of $\geq 1 - \delta/2$ according to Lemma B.1. This completes the proof.

$\square$

Note that $c_T = \widetilde{\mathcal{O}}(R\sqrt{\Gamma_{T\mathbb{B}}})$, which allows us to simplify the regret upper bound into an asymptotic expression:

$$
R_T = \widetilde{\mathcal{O}} \left( e^C R \frac{1}{\sqrt{\mathbb{B}/\lceil \frac{\sigma_{\max}^2}{R^2} \rceil - 1}} \sqrt{T\Gamma_{T\mathbb{B}}} \left(\sqrt{C} + \sqrt{\Gamma_{T\mathbb{B}}}\right) \right).
$$
(23)

## B.1 Simplified Regret Upper Bound for Constant Noise Variance

In the special case where the noise variance is fixed throughout the entire domain, i.e., when $\sigma^2(\boldsymbol{x}) = \sigma_{\text{const}}^2, \forall \boldsymbol{x} \in \mathcal{X}$, we have that $n_t = \lceil \sigma_{\text{const}}^2/R^2 \rceil = n_{\text{const}}$ and $b_t = \lfloor \mathbb{B}/n_{\text{const}} \rfloor = b_0, \forall t \in [T]$. In this case, instead of making use of the lower bound on $b_t$ (i.e., $b_t \geq \mathbb{B}/\lceil \frac{\sigma_{\max}^2}{R^2} \rceil - 1$) as done in the proof of Lemma B.10, we can simply replace $b_t$ with $b_0$ in the proof. As a result, the term of $\frac{1}{\sqrt{\mathbb{B}/\lceil \frac{\sigma_{\max}^2}{R^2} \rceil - 1}}$ in the regret upper bound proved in Lemma B.10 can be simply replaced by $\frac{1}{\sqrt{b_0}}$, and hence the regret upper bound can be further simplified as:

$$
R_T = \widetilde{\mathcal{O}} \left( e^C R \frac{1}{\sqrt{b_0}} \sqrt{T\Gamma_{Tb_0}} \left(\sqrt{C} + \sqrt{\Gamma_{Tb_0}}\right) \right).
$$
(24)

As another special case where $\sigma^2(\boldsymbol{x}) = \sigma_{\text{const}}^2 = R^2$, every input $\boldsymbol{x}$ will be evaluated only once (i.e., $n_{\text{const}} = 1$) and $b_0 = \mathbb{B}$. In this case, our algorithm reduces to the standard batch TS with a batch size of $\mathbb{B}$, and the regret upper bound becomes

$$
R_T = \widetilde{\mathcal{O}} \left( e^C R \frac{1}{\sqrt{\mathbb{B}}} \sqrt{T\Gamma_{T\mathbb{B}}} \left(\sqrt{C} + \sqrt{\Gamma_{T\mathbb{B}}}\right) \right).
$$
(25)

## C  Choice of $R^2$ by Minimizing the Regret Upper Bound in Theorem 3.1

The regret bound in Theorem 3.1 depends on the parameter $R$ through the term $g \triangleq \sqrt{\frac{R^2}{\mathbb{B}/(\sigma_{\max}^2/R^2 + 1) - 1}}$, in which we have replaced the term $\lceil \sigma_{\max}^2/R^2 \rceil$ by $\sigma_{\max}^2/R^2 + 1$ such that the

resulting regret upper bound is still valid and the subsequent derivations become simplified. Taking the derivative of $g^2$ w.r.t. $R^2$ gives us

$$\frac{\mathrm{d}g^2}{\mathrm{d}R^2} = \frac{(\mathbb{B}-1)(R^2)^2 - 2\sigma_{\max}^2 R^2 - \sigma_{\max}^4}{\left[((\mathbb{B}-1)R^2 - \sigma_{\max}^2)\right]^2}. \tag{26}$$

Setting the above derivative to 0, we have that the value of $R^2$ that minimizes $g^2$ and $g$ (hence minimizing the regret upper bound) is obtained at

$$R^2 = \sigma_{\max}^2 \frac{\sqrt{\mathbb{B}}+1}{\mathbb{B}-1}. \tag{27}$$

## D  Upper Bound on Sequential Cumulative Regret

## E  Confidence Bound for the Noise Variance Function

Here, as discussed in Sec. 3.2 in the main text, we leverage the concentration of the Chi-squared distribution to show that we can interpret the unbiased estimate of the noise variance in equation (1) as a noisy observation corrupted by a sub-Gaussian noise. For every queried input, we use the unbiased empirical variance (1) as the observation to update the GP posterior for the noise variance. First of all, when an input $\boldsymbol{x}_t^{[b]}$ is queried, denote the unbiased empirical variance as $\widetilde{\sigma^2}(\boldsymbol{x}_t^{[b]}) = \sigma^2(\boldsymbol{x}_t^{[b]}) + \epsilon'$, and we will show next that $\epsilon'$ is (with high probability) a sub-Gaussian noise. Denote as $n_t$ the number replications that we use to query $\boldsymbol{x}_t^{[b]}$. Since the unbiased empirical variance of a Gaussian random variable follows a Chi-squared distribution, we have that $\widetilde{\sigma^2}(\boldsymbol{x}_t^{[b]}) \in \left[\frac{\sigma^2(\boldsymbol{x}_t^{[b]})\chi_{n_t-1,\alpha/2}^2}{n_t-1}, \frac{\sigma^2(\boldsymbol{x}_t^{[b]})\chi_{n_t-1,1-\alpha/2}^2}{n_t-1}\right]$ with probability of $\geq 1-\alpha$, in which $\chi_{n_t-1,\alpha/2}^2$ ($\chi_{n_t-1,1-\alpha/2}^2$) denotes the $(\alpha/2)^{\text{th}}$-quantile ($(1-\alpha/2)^{\text{th}}$-quantile) of the Chi-square distribution with $n_t - 1$ degrees of freedom. This allows us to show that

$$
\begin{aligned}
\epsilon' &\in \left[\sigma^2(\boldsymbol{x}_t^{[b]})\left(\frac{\chi_{n_t-1,\alpha/2}^2}{n_t-1}-1\right), \sigma^2(\boldsymbol{x}_t^{[b]})\left(\frac{\chi_{n_t-1,1-\alpha/2}^2}{n_t-1}-1\right)\right] \\
&\in \left[\sigma_{\min}^2\left(\frac{\chi_{n_{\min}-1,\alpha/2}^2}{n_{\min}-1}-1\right), \sigma_{\max}^2\left(\frac{\chi_{n_{\min}-1,1-\alpha/2}^2}{n_{\min}-1}-1\right)\right] \triangleq [L_\alpha, U_\alpha],
\end{aligned}
\tag{28}
$$

which holds with probability $\geq 1-\alpha$. $n_{\min}$ is the minimum number of repetitions we impose on every queried $\boldsymbol{x}_t$. Note that the discussion above on the unbiasedness and boundedness of the noise $\epsilon'$ still holds after we negate the noise variance (i.e., $g(\cdot) = -\sigma^2(\cdot)$) to analyze $\widetilde{y}_t^{[b]} = g(\boldsymbol{x}_t^{[b]}) + \epsilon'$. Then the discussion in the main text (Sec. 3.2) follows.

## F  The BTS-RED-Unknown Algorithm

Our BTS-RED-Unknown algorithm is shown in Algorithm 3.

## G  Proof of Theorem 4.1

Our proof here follows a similar structure to the proof in Appendix B, but non-trivial changes need to be made to account for the change of objective function here. Here, for a given $\omega$, we attempt to upper-bound the mean-variance cumulative regret $R_T^{\text{MV}} = \sum_{t=1}^T \min_{b \in [b_t]} [h_\omega(\boldsymbol{x}_\omega^*) - h_\omega(\boldsymbol{x}_t^{[b]})]$, where $h_\omega(\boldsymbol{x}) = \omega f(\boldsymbol{x}) + (1-\omega)g(\boldsymbol{x}) = \omega f(\boldsymbol{x}) - (1-\omega)\sigma^2(\boldsymbol{x})$ and $\boldsymbol{x}_\omega^* \in \arg\max_{\boldsymbol{x} \in \mathcal{X}} [\omega f(\boldsymbol{x}) + (1-\omega)g(\boldsymbol{x})]$. Throughout the analysis here, we assumed a pre-defined $\omega$ and hence omit any dependence on $\omega$ for simplicity. That is, we use $\boldsymbol{x}^*$ in place of $\boldsymbol{x}_\omega^*$ and use $h(\cdot)$ to represent $h_\omega(\cdot)$.

**Algorithm 3** BTS-RED-Unknown.
___
1: **for** $t = 1, 2, \ldots, T$ **do**
2:     $b = 0, n_t^{(0)} = 0$
3:     **while** $\sum_{b'=0}^{b} n_t^{(b')} < \mathbb{B}$ **do**
4:         $b \leftarrow b + 1$
5:         Sample $f_t^{(b)}$ from $\mathcal{GP}(\mu_{t-1}(\cdot), \beta_t^2 \sigma_{t-1}^2(\cdot, \cdot))$
6:         Choose $\boldsymbol{x}_t^{(b)} = \arg\max_{\boldsymbol{x} \in \mathcal{X}} f_t^{(b)}(\boldsymbol{x})$
7:         Choose $n_t^{(b)} = \lceil (-\mu_{t-1}'(\boldsymbol{x}_t^{(b)}) + \beta_t' \sigma_{t-1}'(\boldsymbol{x}_t^{(b)}))/R^2 \rceil$
8:     $b_t = b - 1$
9:     **for** $b \in [b_t]$, query $\boldsymbol{x}_t^{(b)}$ with $n_t^{(b)}$ parallel processes
10:     **for** $b \in [b_t]$, observe $\{y_{t,n}^{(b)}\}_{n \in [n_t^{(b)}]}$, then calculate $y_t^{(b)} = (1/n_t^{(b)}) \sum_{n=1}^{n_t^{(b)}} y_{t,n}^{(b)}$ and $\widetilde{y}_t^{(b)} =$
        $-1/(n_t^{(b)} - 1) \sum_{n=1}^{n_t^{(b)}} \left( y_{t,n}^{(b)} - y_t^{(b)} \right)^2$
11:     Use $\{(\boldsymbol{x}_t^{(b)}, y_t^{(b)})\}_{b \in [b_t]}$ to update posterior of $\mathcal{GP}$
12:     Use $\{(\boldsymbol{x}_t^{(b)}, \widetilde{y}_t^{(b)})\}_{b \in [b_t]}$ to update posterior of $\mathcal{GP}'$
___

Here, same as the main text, we use $g : \mathcal{X} \to \mathbb{R}^-$ to denote the function $-\sigma^2 : \mathcal{X} \to \mathbb{R}^-$. We use $\mu_{t-1}$ and $\sigma_{t-1}$ to denote the GP posterior mean and standard deviation of the GP for $f$ conditioned on all $\tau_{t-1}$ observations up to (and including) iteration $t - 1$, and use $\mu_{t-1}'$ and $\sigma_{t-1}'$ to denote the GP posterior mean and standard deviation of the GP for the negative noise variance $-\sigma^2$. Denote $h_t^{[b]}(\boldsymbol{x}) = \omega f_t^{[b]}(\boldsymbol{x}) + (1 - \omega) g_t^{[b]}(\boldsymbol{x})$, such that $\boldsymbol{x}_t^{[b]} \in \arg\max_{\boldsymbol{x} \in \mathcal{X}} h_t^{[b]}$. Denote by $\mathcal{F}_{t-1}'$ the history of observed pairs of input and empirical noise variance up to iteration $t - 1$.

Define $\beta_t \triangleq B + R\sqrt{2(\Gamma_{\tau_{t-1}} + 1 + \log(3/\delta))}$ and $c_t \triangleq \beta_t(1 + \sqrt{2\log(2\mathbb{B}|\mathcal{X}|t^2)})$. Also define $\beta_t' \triangleq B' + R'\sqrt{2(\Gamma_{\tau_{t-1}}' + 1 + \log(3/\delta))}$ and $c_t' \triangleq \beta_t'(1 + \sqrt{2\log(2\mathbb{B}|\mathcal{X}|t^2)})$.

**Lemma G.1.** *Let $\delta \in (0, 1)$. Define $E^g(t)$ as the event that $|\mu_{t-1}'(\boldsymbol{x}) - g(\boldsymbol{x})| \leq \beta_t' \sigma_{t-1}'(\boldsymbol{x})$ for all $\boldsymbol{x} \in \mathcal{X}$. We have that $\mathbb{P}\left[ E^g(t) \right] \geq 1 - \delta/3$ for all $t \geq 1$.*

The validity of Lemma G.1 follows from the discussion in Sec. 3.2 (i.e., equation (G.1)), after replacing the error probability of $\delta/2$ by $\delta/3$. We also need Lemma B.1 to hold, and since we have replaced the error probability of $\delta/2$ in $\beta_t$ from Lemma B.1 by $\delta/3$ in our definition of $\beta_t$ above, we have that the event $E^f(t)$ in Lemma B.1 holds with probability of $\geq 1 - \delta/3$ here.

**Lemma G.2.** *Define $E^{g_t}(t)$ as the event: $|g_t^{[b]}(\boldsymbol{x}) - \mu_{t-1}'(\boldsymbol{x})| \leq \beta_t' \sqrt{2\log(2\mathbb{B}|\mathcal{X}|t^2)} \sigma_{t-1}'(\boldsymbol{x})$, $\forall b \in [b_t]$. We have that $\mathbb{P}\left[ E^{g_t}(t) | \mathcal{F}_{t-1}' \right] \geq 1 - 1/(2t^2)$ for any possible filtration $\mathcal{F}_{t-1}'$.*

Similarly, we will also need Lemma B.2 to hold, but replace the error probability of $1/t^2$ by $1/(2t^2)$. Of note, we have also correspondingly changed the value of $c_t$ from Appendix B by replacing $t^2$ by $2t^2$ in our definition of $c_t$ above.

The definition of saturated points also needs to be modified:

**Definition G.3.** Define the set of saturated points at iteration $t$ as

$$S_t' = \{\boldsymbol{x} \in \mathcal{X} : \Delta(\boldsymbol{x}) > \omega c_t \sigma_{t-1}(\boldsymbol{x}) + (1 - \omega) c_t' \sigma_{t-1}'(\boldsymbol{x})\},$$

in which $\Delta(\boldsymbol{x}) = h(\boldsymbol{x}^*) - h(\boldsymbol{x})$ and $\boldsymbol{x}^* \in \arg\max_{\boldsymbol{x} \in \mathcal{X}} h(\boldsymbol{x})$.

The following lemma is a counterpart to Lemma B.4 in Appendix B, and the proof here makes use of the same techniques.

**Lemma G.4.** *For any $\mathcal{F}_{t-1}'$, conditioned on the events $E^g(t)$, we have that $\forall \boldsymbol{x} \in \mathcal{X}, b \in [b_t]$,*

$$\mathbb{P}\left( g_t^{[b]}(\boldsymbol{x}) > g(\boldsymbol{x}) | \mathcal{F}_{t-1}' \right) \geq p, \tag{29}$$

*in which $p = \frac{1}{4e\sqrt{\pi}}$.*

*Proof.* For any $b \in [b_t]$, we have that

$$\mathbb{P}\left(g_t^{[b]}(\boldsymbol{x}) > g(\boldsymbol{x})|\mathcal{F}'_{t-1}\right) = \mathbb{P}\left(\frac{g_t^{[b]}(\boldsymbol{x}) - \mu'_{t-1}(\boldsymbol{x})}{\beta'_t \sigma'_{t-1}(\boldsymbol{x})} > \frac{g(\boldsymbol{x}) - \mu'_{t-1}(\boldsymbol{x})}{\beta'_t \sigma'_{t-1}(\boldsymbol{x})}\Big|\mathcal{F}'_{t-1}\right)$$

$$\geq \mathbb{P}\left(\frac{g_t^{[b]}(\boldsymbol{x}) - \mu'_{t-1}(\boldsymbol{x})}{\beta'_t \sigma'_{t-1}(\boldsymbol{x})} > \frac{|g(\boldsymbol{x}) - \mu'_{t-1}(\boldsymbol{x})|}{\beta'_t \sigma'_{t-1}(\boldsymbol{x})}\Big|\mathcal{F}'_{t-1}\right) \tag{30}$$

$$\geq \mathbb{P}\left(\frac{g_t^{[b]}(\boldsymbol{x}) - \mu'_{t-1}(\boldsymbol{x})}{\beta'_t \sigma'_{t-1}(\boldsymbol{x})} > 1\Big|\mathcal{F}'_{t-1}\right)$$

$$\geq \frac{e^{-1}}{4\sqrt{\pi}}.$$

$\square$

The next Lemma is the counterpart to Lemma B.5 in Appendix B, but additional challenges need to be carefully handled here.

**Lemma G.5.** *For any $\mathcal{F}_{t-1}$ and $\mathcal{F}'_{t-1}$, conditioned on the events $E^f(t)$ and $E^g(t)$, we have that*

$$\mathbb{P}\left(\boldsymbol{x}_t^{[b]} \in \mathcal{X} \setminus S'_t|\mathcal{F}_{t-1}, \mathcal{F}'_{t-1}\right) \geq p^2 - 1/t^2, \qquad \forall b \in [b_t].$$

*Proof.* For every $b \in [b_t]$,

$$\mathbb{P}\left(\boldsymbol{x}_t^{[b]} \in \mathcal{X} \setminus S'_t|\mathcal{F}_{t-1}, \mathcal{F}'_{t-1}\right) \geq \mathbb{P}\left(h_t^{[b]}(\boldsymbol{x}^*) > h_t^{[b]}(\boldsymbol{x}), \forall \boldsymbol{x} \in S_t|\mathcal{F}_{t-1}, \mathcal{F}'_{t-1}\right), \tag{31}$$

which holds $\forall b \in [b_t]$. This inequality follows from noting that $\boldsymbol{x}^*$ is always unsaturated according to Definition G.3, and that $\boldsymbol{x}_t^{[b]} = \arg\max_{\boldsymbol{x} \in \mathcal{X}} h_t^{[b]}(\boldsymbol{x}) = \arg\max_{\boldsymbol{x} \in \mathcal{X}}(\omega f_t^{[b]}(\boldsymbol{x}) + (1 - \omega)g_t^{[b]}(\boldsymbol{x}))$.

Next, we assume that the events $E^f(t)$, $E^{f_t}(t)$, $E^g(t)$ and $E^{g_t}(t)$ all hold, which allows us to derive an upper bound on $h_t^{[b]}(\boldsymbol{x})$ for all $\boldsymbol{x} \in S'_t$ and for all $b \in [b_t]$:

$$h_t^{[b]}(\boldsymbol{x}) = \omega f_t^{[b]}(\boldsymbol{x}) + (1 - \omega)g_t^{[b]}(\boldsymbol{x})$$

$$\leq \omega \left(f(\boldsymbol{x}) + c_t \sigma_{t-1}(\boldsymbol{x})\right) + (1 - \omega)\left(g(\boldsymbol{x}) + c'_t \sigma'_{t-1}(\boldsymbol{x})\right)$$

$$= \omega f(\boldsymbol{x}) + (1 - \omega)g(\boldsymbol{x}) + \omega c_t \sigma_{t-1}(\boldsymbol{x}) + (1 - \omega)c'_t \sigma'_{t-1}(\boldsymbol{x}) \tag{32}$$

$$\leq h(\boldsymbol{x}) + \Delta(\boldsymbol{x})$$

$$= h(\boldsymbol{x}) + h(\boldsymbol{x}^*) - h(\boldsymbol{x}) = h(\boldsymbol{x}^*),$$

where the first inequality follows from Lemmas B.1, B.2, G.1 and G.2, and the second inequality makes use of the definition of $h(\boldsymbol{x})$ as well as Definition G.3.

Therefore, (32) implies that for every $b \in [b_t]$, if the events $E^f(t)$, $E^{f_t}(t)$, $E^g(t)$ and $E^{g_t}(t)$ all hold, then we have that

$$\mathbb{P}\left(h_t^{[b]}(\boldsymbol{x}^*) > h_t^{[b]}(\boldsymbol{x}), \forall \boldsymbol{x} \in S_t|\mathcal{F}_{t-1}, \mathcal{F}'_{t-1}\right) \geq \mathbb{P}\left(h_t^{[b]}(\boldsymbol{x}^*) > h(\boldsymbol{x}^*)|\mathcal{F}_{t-1}, \mathcal{F}'_{t-1}\right)$$

$$= \mathbb{P}\left(\omega f_t^{[b]}(\boldsymbol{x}^*) + (1 - \omega)g_t^{[b]}(\boldsymbol{x}^*) > \omega f(\boldsymbol{x}^*) + (1 - \omega)g(\boldsymbol{x}^*)|\mathcal{F}_{t-1}, \mathcal{F}'_{t-1}\right)$$

$$\geq \mathbb{P}\left(f_t^{[b]}(\boldsymbol{x}^*) > f(\boldsymbol{x}^*) \text{ and } g_t^{[b]}(\boldsymbol{x}^*) > g(\boldsymbol{x}^*)|\mathcal{F}_{t-1}, \mathcal{F}'_{t-1}\right)$$

$$\geq p^2. \tag{33}$$

The second inequality results since the event in the third line implies the event in the line above, and the last inequality follows from Lemmas B.4 and G.4.

Next, for every $b \in [b_t]$, we can show that

$$\mathbb{P}\left(\boldsymbol{x}_t^{[b]} \in \mathcal{X} \setminus S_t' | \mathcal{F}_{t-1}, \mathcal{F}_{t-1}'\right) \geq \mathbb{P}\left(h_t^{[b]}(\boldsymbol{x}^*) > h_t^{[b]}(\boldsymbol{x}), \forall \boldsymbol{x} \in S_t' | \mathcal{F}_{t-1}, \mathcal{F}_{t-1}'\right)$$

$$\geq \mathbb{P}\left(h_t^{[b]}(\boldsymbol{x}^*) > h(\boldsymbol{x}^*) | \mathcal{F}_{t-1}, \mathcal{F}_{t-1}\right) - \mathbb{P}\left(\overline{E^{f_t}(t)} \cup \overline{E^{g_t}(t)} | \mathcal{F}_{t-1}, \mathcal{F}_{t-1}'\right)$$

$$\geq p^2 - 1/t^2,$$

(34)

which holds for all $b \in [b_t]$. The second inequality follows from considering the following two events separately: $E^{f_t}(t) \cap E^{g_t}(t)$ and $\overline{E^{f_t}(t) \cap E^{g_t}(t)}$, and the last inequality follows since $\mathbb{P}\left(\overline{E^{f_t}(t)} \cup \overline{E^{g_t}(t)} | \mathcal{F}_{t-1}, \mathcal{F}_{t-1}'\right) \leq \mathbb{P}\left(\overline{E^{f_t}(t)} | \mathcal{F}_{t-1}, \mathcal{F}_{t-1}'\right) + \mathbb{P}\left(\overline{E^{g_t}(t)} | \mathcal{F}_{t-1}, \mathcal{F}_{t-1}'\right) = 1/(2t^2) + 1/(2t^2) = 1/t^2$. $\qquad \square$

Next, we will need the following Lemma which is a counterpart to Lemma B.6.

**Lemma G.6.** *Define* $C_2 = \frac{2}{\log(1+\lambda^{-1})}$. *Denote all* $\tau_{t-1}$ *observed empirical means from iterations (batches)* 1 *to* $t-1$ *as* $\boldsymbol{y}_{1:t-1}$, *and the* $b_t$ *observed empirical means in the* $t^{th}$ *batch as* $\boldsymbol{y}_t$. *Also denote all* $\tau_{t-1}$ *observed noise variances from iterations (batches)* 1 *to* $t-1$ *as* $\boldsymbol{y}_{1:t-1}'$, *and the* $b_t$ *observed noise variances in the* $t^{th}$ *batch as* $\boldsymbol{y}_t'$. *Choose* $C$ *as an absolute constant such that* $\max_{A \subset \mathcal{X}, |A| \leq \mathbb{B}} \mathbb{I}\left(f; \boldsymbol{y}_A | \boldsymbol{y}_{1:t-1}\right) \leq C, \forall t \geq 1$ *and* $\max_{A \subset \mathcal{X}, |A| \leq \mathbb{B}} \mathbb{I}\left(g; \boldsymbol{y}_A' | \boldsymbol{y}_{1:t-1}'\right) \leq C, \forall t \geq 1$. *Then we have that*

$$\sum_{b=1}^{b_t} \sigma_{t-1}(\boldsymbol{x}_t^{[b]}) \leq e^C \sqrt{C_2 b_t \mathbb{I}(f; \boldsymbol{y}_t | \boldsymbol{y}_{1:t-1})},$$

*and*

$$\sum_{b=1}^{b_t} \sigma_{t-1}'(\boldsymbol{x}_t^{[b]}) \leq e^C \sqrt{C_2 b_t \mathbb{I}(g; \boldsymbol{y}_t' | \boldsymbol{y}_{1:t-1}')}.$$

*Proof.* Note that we have assumed that both $f$ and $g$ are associated with the SE kernels $k$ and $k'$, respectively. Denote the length scales for $k$ and $k'$ by $\theta$ and $\theta'$ respectively. Note that the maximum information gain is decreasing in the length scale, i.e., a smaller length scale leads to a larger maximum information gain [1]. Therefore, we can run the initialization stage via uncertainty sampling to observe $\boldsymbol{y}_{\text{init}}$ using the kernel with a smaller length scale.

For example, if $\theta < \theta'$, we use the kernel $k$ to run the uncertainty sampling algorithm for $T_{\text{init}}$ iterations to collect the initial set of inputs $D_{\text{init}}$, such that we can guarantee that

$$\max_{A \subset \mathcal{X}, |A| \leq \mathbb{B}} \mathbb{I}\left(f; \boldsymbol{y}_A | \boldsymbol{y}_{1:t-1}\right) \leq \max_{A \subset \mathcal{X}, |A| \leq \mathbb{B}} \mathbb{I}\left(f; \boldsymbol{y}_A | \boldsymbol{y}_{\text{init}}\right) \leq C, \forall t \geq 1, \qquad (35)$$

where the first inequality follows from the submodularity of conditional information gain, and the second inequality is a consequence of Lemma 4 of [19]. Note that given the same set of initial inputs $D_{\text{init}}$, the maximum conditional information gains $\max_{A \subset \mathcal{X}, |A| \leq \mathbb{B}} \mathbb{I}\left(f; \boldsymbol{y}_A | \boldsymbol{y}_{\text{init}}\right)$ and $\max_{A \subset \mathcal{X}, |A| \leq \mathbb{B}} \mathbb{I}\left(g; \boldsymbol{y}_A' | \boldsymbol{y}_{\text{init}}'\right)$ differ by only the lengthscales of the kernels $k$ and $k'$, denoted as $\theta$ and $\theta'$, respectively. Therefore, since we have assumed that $\theta < \theta'$ in the discussion here, we have that

$$\max_{A \subset \mathcal{X}, |A| \leq \mathbb{B}} \mathbb{I}\left(g; \boldsymbol{y}_A' | \boldsymbol{y}_{\text{init}}'\right) < \max_{A \subset \mathcal{X}, |A| \leq \mathbb{B}} \mathbb{I}\left(f; \boldsymbol{y}_A | \boldsymbol{y}_{\text{init}}\right). \qquad (36)$$

This further tells us that

$$\max_{A \subset \mathcal{X}, |A| \leq \mathbb{B}} \mathbb{I}\left(g; \boldsymbol{y}_A' | \boldsymbol{y}_{1:t-1}'\right) \leq \max_{A \subset \mathcal{X}, |A| \leq \mathbb{B}} \mathbb{I}\left(g; \boldsymbol{y}_A' | \boldsymbol{y}_{\text{init}}'\right) \leq \max_{A \subset \mathcal{X}, |A| \leq \mathbb{B}} \mathbb{I}\left(f; \boldsymbol{y}_A | \boldsymbol{y}_{\text{init}}\right) \leq C, \forall t \geq 1. \qquad (37)$$

Subsequently, the proof is completed by applying the proof techniques of Lemma B.6 to $\sigma_{t-1}$ and $\sigma_{t-1}'$ separately. $\qquad \square$

**Lemma G.7.** *Define $\widetilde{B} = \omega B + (1-\omega)B'$. For any filtrations $\mathcal{F}_{t-1}$ and $\mathcal{F}'_{t-1}$, conditioned on the events $E^f(t)$ and $E^g(t)$, we have that*

$$\mathbb{E}\left[\min_{b \in [b_t]} r_t^{[b]} \Big| \mathcal{F}_{t-1}\right] \le e^C \left(1 + \frac{28}{p^2}\right) \left[\omega c_t \mathbb{E}\Big[\frac{1}{b_t}\sqrt{C_2 b_t \mathbb{I}\left(f; \boldsymbol{y}_t | \boldsymbol{y}_{1:t-1}\right)} | \mathcal{F}_{t-1}, \mathcal{F}'_{t-1}\Big]\right.$$

$$\left. + (1-\omega)c'_t \mathbb{E}\Big[\frac{1}{b_t}\sqrt{C_2 b_t \mathbb{I}\left(g; \boldsymbol{y}'_t | \boldsymbol{y}'_{1:t-1}\right)} | \mathcal{F}_{t-1}, \mathcal{F}'_{t-1}\Big]\right] + \frac{2\widetilde{B}}{t^2},$$

*in which $r_t^{[b]} = h(\boldsymbol{x}^*) - h(\boldsymbol{x}_t^{[b]})$.*

*Proof.* To begin with, we define $\overline{\boldsymbol{x}}_t$ as the unsaturated input at iteration $t$ with the smallest weighted posterior standard deviation:

$$\overline{\boldsymbol{x}}_t = \arg\min_{\boldsymbol{x} \in \mathcal{X} \setminus S'_t} \left(\omega c_t \sigma_{t-1}(\boldsymbol{x}) + (1-\omega)c'_t \sigma'_{t-1}(\boldsymbol{x})\right). \tag{38}$$

Following this definition, if both $E^f(t)$ and $E^g(t)$ hold $\forall b \in [b_t]$, then we have that

$$\mathbb{E}\left[\omega c_t \sigma_{t-1}(\boldsymbol{x}_t^{[b]}) + (1-\omega)c'_t \sigma'_{t-1}(\boldsymbol{x}_t^{[b]}) | \mathcal{F}_{t-1}, \mathcal{F}'_{t-1}\right]$$

$$\ge \mathbb{E}\left[\omega c_t \sigma_{t-1}(\boldsymbol{x}_t^{[b]}) + (1-\omega)c'_t \sigma'_{t-1}(\boldsymbol{x}_t^{[b]}) | \mathcal{F}_{t-1}, \mathcal{F}'_{t-1}, \boldsymbol{x}_t^{[b]} \in \mathcal{X} \setminus S'_t\right] \mathbb{P}\left(\boldsymbol{x}_t^{[b]} \in \mathcal{X} \setminus S'_t | \mathcal{F}_{t-1}, \mathcal{F}'_{t-1}\right)$$

$$\ge \left[\omega c_t \sigma_{t-1}(\overline{\boldsymbol{x}}_t) + (1-\omega)c'_t \sigma'_{t-1}(\overline{\boldsymbol{x}}_t)\right](p^2 - 1/t^2) \tag{39}$$

Now we condition on all events $E^f(t)$, $E^{f_t}(t)$, $E^g(t)$ and $E^{g_t}(t)$, and analyze the instantaneous regret as:

$$\min_{b \in [b_t]} r_t^{[b]} \le \frac{1}{b_t}\sum_{b=1}^{b_t} r_t^{[b]} = \frac{1}{b_t}\sum_{b=1}^{b_t} \Delta(\boldsymbol{x}_t^{[b]}) = \frac{1}{b_t}\sum_{b=1}^{b_t}\left[h(\boldsymbol{x}^*) - h(\overline{\boldsymbol{x}}_t) + h(\overline{\boldsymbol{x}}_t) - h(\boldsymbol{x}_t^{[b]})\right]$$

$$\le \frac{1}{b_t}\sum_{b=1}^{b_t}\left[\Delta(\overline{\boldsymbol{x}}_t) + h_t^{[b]}(\overline{\boldsymbol{x}}_t) + \omega c_t \sigma_{t-1}(\overline{\boldsymbol{x}}_t) + (1-\omega)c'_t \sigma'_{t-1}(\overline{\boldsymbol{x}}_t)\right.$$

$$\left. - h_t^{[b]}(\boldsymbol{x}_t^{[b]}) + \omega c_t \sigma_{t-1}(\boldsymbol{x}_t^{[b]}) + (1-\omega)c'_t \sigma'_{t-1}(\boldsymbol{x}_t^{[b]})\right]$$

$$\le \frac{1}{b_t}\sum_{b=1}^{b_t}\left[\omega c_t \sigma_{t-1}(\overline{\boldsymbol{x}}_t) + (1-\omega)c'_t \sigma'_{t-1}(\overline{\boldsymbol{x}}_t)\right.$$

$$+ h_t^{[b]}(\overline{\boldsymbol{x}}_t) + \omega c_t \sigma_{t-1}(\overline{\boldsymbol{x}}_t) + (1-\omega)c'_t \sigma'_{t-1}(\overline{\boldsymbol{x}}_t)$$

$$\left. - h_t^{[b]}(\boldsymbol{x}_t^{[b]}) + \omega c_t \sigma_{t-1}(\boldsymbol{x}_t^{[b]}) + (1-\omega)c'_t \sigma'_{t-1}(\boldsymbol{x}_t^{[b]})\right]$$

$$\le \frac{1}{b_t}\sum_{b=1}^{b_t}\left[\omega c_t\left(2\sigma_{t-1}(\overline{\boldsymbol{x}}_t) + \sigma_{t-1}(\boldsymbol{x}_t^{[b]})\right) + (1-\omega)c'_t\left(2\sigma'_{t-1}(\overline{\boldsymbol{x}}_t) + \sigma'_{t-1}(\boldsymbol{x}_t^{[b]})\right)\right.$$

$$\left. + h_t^{[b]}(\overline{\boldsymbol{x}}_t) - h_t^{[b]}(\boldsymbol{x}_t^{[b]})\right]$$

$$\le \frac{1}{b_t}\sum_{b=1}^{b_t}\left[\omega c_t\left(2\sigma_{t-1}(\overline{\boldsymbol{x}}_t) + \sigma_{t-1}(\boldsymbol{x}_t^{[b]})\right) + (1-\omega)c'_t\left(2\sigma'_{t-1}(\overline{\boldsymbol{x}}_t) + \sigma'_{t-1}(\boldsymbol{x}_t^{[b]})\right)\right]$$

$$= \frac{1}{b_t}\sum_{b=1}^{b_t}\left[2\left(\omega c_t \sigma_{t-1}(\overline{\boldsymbol{x}}_t) + (1-\omega)c'_t \sigma'_{t-1}(\overline{\boldsymbol{x}}_t)\right) + \omega c_t \sigma_{t-1}(\boldsymbol{x}_t^{[b]}) + (1-\omega)c'_t \sigma'_{t-1}(\boldsymbol{x}_t^{[b]})\right], \tag{40}$$

in which the second inequality follows from Lemmas B.1, B.2, G.1 and G.2, the third inequality results from Definition G.3 and the fact that $\overline{\boldsymbol{x}}_t$ is unsaturated, and the last inequality follows from the way in which $\boldsymbol{x}_t^{[b]}$ is selected: $\boldsymbol{x}_t^{[b]} = \arg\max_{\boldsymbol{x} \in \mathcal{X}} h_t^{[b]}(\boldsymbol{x})$.

$$\mathbb{E}\left[\min_{b\in[b_t]} r_t^{[b]}\Big|\mathcal{F}_{t-1},\mathcal{F}'_{t-1}\right] \leq \mathbb{E}\left[\frac{1}{b_t}\sum_{b=1}^{b_t}\left[2\left(\omega c_t\sigma_{t-1}(\overline{\boldsymbol{x}}_t)+(1-\omega)c'_t\sigma'_{t-1}(\overline{\boldsymbol{x}}_t)\right)\right.\right.$$

$$\left.\left.+\omega c_t\sigma_{t-1}(\boldsymbol{x}_t^{[b]})+(1-\omega)c'_t\sigma'_{t-1}(\boldsymbol{x}_t^{[b]})\right]\Big|\mathcal{F}_{t-1},\mathcal{F}'_{t-1}\right] + 2\widetilde{B}\mathbb{P}\left[\overline{E^{f_t}(t)}\cup\overline{E^{g_t}(t)}|\mathcal{F}_{t-1},\mathcal{F}'_{t-1}\right]$$

$$\leq \mathbb{E}\left[\frac{1}{b_t}\sum_{b=1}^{b_t}\left[\frac{2}{p^2-1/t^2}\left(\omega c_t\sigma_{t-1}(\boldsymbol{x}_t^{[b]})+(1-\omega)c'_t\sigma'_{t-1}(\boldsymbol{x}_t^{[b]})\right)\right.\right.$$

$$\left.\left.+\omega c_t\sigma_{t-1}(\boldsymbol{x}_t^{[b]})+(1-\omega)c'_t\sigma'_{t-1}(\boldsymbol{x}_t^{[b]})\right]\Big|\mathcal{F}_{t-1},\mathcal{F}'_{t-1}\right] + 2\widetilde{B}\frac{1}{t^2}$$

$$\leq \omega c_t\left(1+\frac{28}{p^2}\right)\mathbb{E}\left[\frac{1}{b_t}\sum_{b=1}^{b_t}\sigma_{t-1}(\boldsymbol{x}_t^{[b]})|\mathcal{F}_{t-1},\mathcal{F}'_{t-1}\right]$$

$$+(1-\omega)c'_t\left(1+\frac{28}{p^2}\right)\mathbb{E}\left[\frac{1}{b_t}\sum_{b=1}^{b_t}\sigma'_{t-1}(\boldsymbol{x}_t^{[b]})|\mathcal{F}_{t-1},\mathcal{F}'_{t-1}\right] + \frac{2\widetilde{B}}{t^2}$$

$$\leq \omega c_t e^C\left(1+\frac{28}{p^2}\right)\mathbb{E}\left[\frac{1}{b_t}\sqrt{C_2 b_t\mathbb{I}\left(f;\boldsymbol{y}_t|\boldsymbol{y}_{1:t-1}\right)}|\mathcal{F}_{t-1},\mathcal{F}'_{t-1}\right]$$

$$+(1-\omega)c'_t e^C\left(1+\frac{28}{p^2}\right)\mathbb{E}\left[\frac{1}{b_t}\sqrt{C_2 b_t\mathbb{I}\left(g;\boldsymbol{y}'_t|\boldsymbol{y}'_{1:t-1}\right)}|\mathcal{F}_{t-1},\mathcal{F}'_{t-1}\right] + \frac{2\widetilde{B}}{t^2}$$

$$= e^C\left(1+\frac{28}{p^2}\right)\left[\omega c_t\mathbb{E}\left[\frac{1}{b_t}\sqrt{C_2 b_t\mathbb{I}\left(f;\boldsymbol{y}_t|\boldsymbol{y}_{1:t-1}\right)}|\mathcal{F}_{t-1},\mathcal{F}'_{t-1}\right]\right.$$

$$\left.+(1-\omega)c'_t\mathbb{E}\left[\frac{1}{b_t}\sqrt{C_2 b_t\mathbb{I}\left(g;\boldsymbol{y}'_t|\boldsymbol{y}'_{1:t-1}\right)}|\mathcal{F}_{t-1},\mathcal{F}'_{t-1}\right]\right] + \frac{2\widetilde{B}}{t^2}.$$

$$(41)$$

The second inequality follows from equation (39), the third inequality follows since $1/(p^2-1/t^2)\leq 14/p^2$, and the last inequality makes use of Lemma G.6.

$\square$

**Definition G.8.** Define $Y_0 = 0$, and for all $t = 1,\ldots,T$,

$$\overline{r}_t = \mathbb{I}\{E^f(t)\cap E^g(t)\}\min_{b\in[b_t]} r_t^{[b]},$$

$$X_t = \overline{r}_t - e^C\left(1+\frac{28}{p^2}\right)\left[\omega c_t\frac{1}{b_t}\sqrt{C_2 b_t\mathbb{I}\left(f;\boldsymbol{y}_t|\boldsymbol{y}_{1:t-1}\right)}+(1-\omega)c'_t\frac{1}{b_t}\sqrt{C_2 b_t\mathbb{I}\left(g;\boldsymbol{y}'_t|\boldsymbol{y}'_{1:t-1}\right)}\right] - \frac{2\widetilde{B}}{t^2}$$

$$Y_t = \sum_{s=1}^{t} X_s.$$

Following the proof of Lemma B.9, we can easily show that $(Y_t : t = 0,\ldots,T)$ is a super-martingale. Now we are finally ready to prove an upper bound on the batch cumulative regret of our Mean-Var-BTS-RED:

**Lemma G.9.** *Define* $C_0 \triangleq \frac{1}{\mathbb{B}/\lceil\frac{\sigma_{\max}^2}{R^2}\rceil-1}$. *Given* $\delta\in(0,1)$, *then with probability of at least* $1-\delta$,

$$R_T^{MV} \leq e^C\left(1+\frac{28}{p^2}\right)\sqrt{C_2 C_0}\sqrt{T}\left[\omega c_T\sqrt{\Gamma_{T\mathbb{B}}}+(1-\omega)c'_T\sqrt{\Gamma'_{T\mathbb{B}}}\right]$$

$$+\frac{\widetilde{B}\pi^2}{3}+\left(6\widetilde{B}+e^C\sqrt{C}\left(1+\frac{28}{p^2}\right)\sqrt{C_2 C_0}\left[\omega c_T+(1-\omega)c'_T\right]\right)\sqrt{2\log(4/\delta)T},$$

*in which* $\Gamma_{T\mathbb{B}}$ *is the maximum information gain about* $f$ *obtained from any set of* $T\mathbb{B}$ *observations, and* $\Gamma'_{T\mathbb{B}}$ *is the maximum information gain about* $g$ *obtained from any set of* $T\mathbb{B}$ *observations.*

*Proof.* To begin with, let's derive a lower bound on $b_t$. Firstly, note that $n_t \leq \lceil \frac{\sigma_{\max}^2}{R^2} \rceil$. This implies that $b_t \geq \mathbb{B}/\lceil \frac{\sigma_{\max}^2}{R^2} \rceil - 1$.

$$
\begin{aligned}
|Y_t - Y_{t-1}| &= |X_t| \\
&= |\bar{r}_t| + e^C \left(1 + \frac{28}{p^2}\right) \left[ \omega c_t \frac{1}{b_t} \sqrt{C_2 b_t \mathbb{I}\left(f; \boldsymbol{y}_t | \boldsymbol{y}_{1:t-1}\right)} \right. \\
&\quad \left. + (1-\omega)c'_t \frac{1}{b_t} \sqrt{C_2 b_t \mathbb{I}\left(g; \boldsymbol{y}'_t | \boldsymbol{y}'_{1:t-1}\right)} \right] + \frac{2\widetilde{B}}{t^2} \\
&\leq 2\widetilde{B} + e^C \left(1 + \frac{28}{p^2}\right) \left[ \omega c_t \sqrt{\frac{C_2 C}{b_t}} + (1-\omega)c'_t \sqrt{\frac{C_2 C}{b_t}} \right] + 2\widetilde{B} \\
&= 4\widetilde{B} + e^C \left(1 + \frac{28}{p^2}\right) \left[ \omega c_t \sqrt{\frac{C_2 C}{b_t}} + (1-\omega)c'_t \sqrt{\frac{C_2 C}{b_t}} \right].
\end{aligned}
\tag{42}
$$

$$
\sum_{t=1}^{T} \sqrt{\mathbb{I}\left(f; \boldsymbol{y}_t | \boldsymbol{y}_{1:t-1}\right)} \leq \sqrt{T \sum_{t=1}^{T} \mathbb{I}\left(f; \boldsymbol{y}_t | \boldsymbol{y}_{1:t-1}\right)} = \sqrt{T \mathbb{I}\left(f; \boldsymbol{y}_{1:T}\right)} \leq \sqrt{T \Gamma_{T\mathbb{B}}}.
\tag{43}
$$

Applying the Azuma-Hoeffding's inequality using an error probability of $\delta/3$ leads to

$$
\begin{aligned}
\sum_{t=1}^{T} \bar{r}_t &\leq e^C \left(1 + \frac{28}{p^2}\right) \left[ \omega \sum_{t=1}^{T} c_t \sqrt{\frac{1}{b_t} C_2 \mathbb{I}\left(f; \boldsymbol{y}_t | \boldsymbol{y}_{1:t-1}\right)} + (1-\omega) \sum_{t=1}^{T} c'_t \sqrt{\frac{1}{b_t} C_2 \mathbb{I}\left(g; \boldsymbol{y}'_t | \boldsymbol{y}'_{1:t-1}\right)} \right] \\
&\quad + \sum_{t=1}^{T} \frac{2\widetilde{B}}{t^2} + \sqrt{2\log(\frac{3}{\delta}) \sum_{t=1}^{T} \left(4\widetilde{B} + e^C \left(1 + \frac{28}{p^2}\right) \left[ \omega c_t \sqrt{\frac{C_2 C}{b_t}} + (1-\omega)c'_t \sqrt{\frac{C_2 C}{b_t}} \right]\right)^2} \\
&\leq e^C \left(1 + \frac{28}{p^2}\right) \sqrt{\frac{C_2}{\mathbb{B}/\lceil \frac{\sigma_{\max}^2}{R^2} \rceil - 1}} \sqrt{T} \left[ \omega c_T \sqrt{\Gamma_{T\mathbb{B}}} + (1-\omega)c'_T \sqrt{\Gamma'_{T,\mathbb{B}}} \right] \\
&\quad + \frac{\widetilde{B}\pi^2}{3} + \left(4\widetilde{B} + e^C \sqrt{C} \left(1 + \frac{28}{p^2}\right) \sqrt{\frac{C_2}{\mathbb{B}/\lceil \frac{\sigma_{\max}^2}{R^2} \rceil - 1}} \left[ \omega c_T + (1-\omega)c'_T \right]\right) \sqrt{2\log(3/\delta)T} \\
&= e^C \left(1 + \frac{28}{p^2}\right) \sqrt{C_2 C_0} \sqrt{T} \left[ \omega c_T \sqrt{\Gamma_{T\mathbb{B}}} + (1-\omega)c'_T \sqrt{\Gamma'_{T\mathbb{B}}} \right] \\
&\quad + \frac{\widetilde{B}\pi^2}{3} + \left(4\widetilde{B} + e^C \sqrt{C} \left(1 + \frac{28}{p^2}\right) \sqrt{C_2 C_0} \left[ \omega c_T + (1-\omega)c'_T \right]\right) \sqrt{2\log(3/\delta)T}.
\end{aligned}
\tag{44}
$$

The second inequality makes use of equation (43) and the lower bound on $b_t$: $b_t \geq \mathbb{B}/\lceil \frac{\sigma_{\max}^2}{R^2} \rceil - 1$. Now, note that $\bar{r}_t = r_t, \forall t \geq 1$ with probability of $\geq 1 - \delta/3 - \delta/3$ because both events $E^f(t)$ and $E^g(t)$ hold with probability of $\geq 1 - \delta/3$ respectively. As a result, taking into account the error probability of $\delta/3$ from the Azuma-Hoeffding's inequality, the regret upper bound holds with probability of $\geq 1 - \delta/3 - \delta/3 - \delta/3 = 1 - \delta$. $\qquad \square$

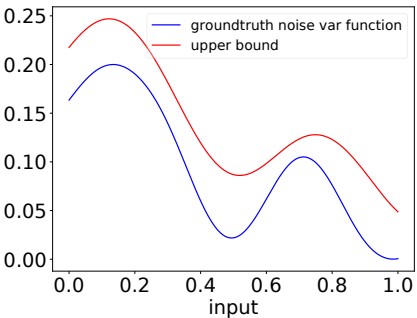

Figure 4: Groundtruth noise variance function and an estimated upper bound (Sec. 3.2).

Finally, we can simplify the regret upper bound from Lemma G.9 into asymptotic notation:

$$R_T^{\text{MV}} = \widetilde{\mathcal{O}} \left( e^C \frac{1}{\sqrt{\mathbb{B}/\lceil \frac{\sigma_{\max}^2}{R^2} \rceil - 1}} \sqrt{T} \left[ \omega R \sqrt{\Gamma_{T\mathbb{B}}} (\sqrt{\Gamma_{T\mathbb{B}}} + \sqrt{C}) + (1-\omega) R' \sqrt{\Gamma'_{T\mathbb{B}}} (\sqrt{\Gamma'_{T\mathbb{B}}} + \sqrt{C}) \right] \right) \tag{45}$$

## H More Experimental Details

Since the theoretical value of $\beta_t$ and $\beta_t'$ is usually too conservative [41, 3], we follow the common practice and set them to a constant: $\beta_t = \beta_t' = 1$. In every experiment, we use the same set of initial inputs selected via random search for all methods to ensure a fair comparison. For all methods based on TS (including all our methods), i.e., all methods which require sampling functions from the GP posterior (e.g., line 5 of Algo. 1, line 5 of Algo. 3 and line 5 of Algo. 2), we follow the common practice and sample the functions from the GP posterior through random Fourier features approximation [39, 46, 34]. Again following the common practice in BO, we optimize the GP hyperparameters by maximizing the marginal likelihood after every 10 iterations. Our experiments are run on a computer server with 128 CPUs, with the AMD EPYC 7543 32-Core Processor. The server has 8 NVIDIA GeForce RTX 3080 GPUs.

### H.1 Synthetic Experiments

In the synthetic experiments, for both mean and mean-variance optimization, we firstly use a sampled function from a GP with the SE kernel (lengthscale= 0.04) as the objective function $f$ (normalized into the range $[0, 1]$), and then sample another function from a GP with the SE kernel (lengthscale= 0.15) as the noise variance function $\sigma^2$ (normalized into the range $[0.0001, 0.2]$).

Fig. 4 shows an example of the upper bound $U_t^{\sigma^2}(\boldsymbol{x}) = -\mu_{t-1}'(\boldsymbol{x}) + \beta_t' \sigma_{t-1}'(\boldsymbol{x})$ constructed by our BTS-RED-Unknown (Sec. 3.2), which is an effective approximation of the groundtruth $\sigma^2(\cdot)$.

Here we also use the synthetic experiments to explore the impact of the uncertainty sampling (US) initialization, which is required by our theoretical results (Sec. 3.1.2), affects the empirical performance of our algorithms. The results in Fig. 8 show that using US and random search as the initialization method lead to similar performances. This provides an empirical justification for our choice of using the simpler random search as the initialization method in our main experiments.

### H.2 Real-world Experiments on Plant Growth

As we have mentioned in the main text (Sec. 5.2), we perform real-world experiments using the input conditions from a regular a 2-D grid within the 2-D domain. Specifically, the 2-D grid is constructed using the pH values of $\{2.5, 3.0, 3.5, 4.0, 4.5, 5.5, 6.5\}$ and NH3 concentrations of $\{0, 1, 5, 10, 15, 20, 25, 30\}$ ($\times 1000$ uM). In other words, the size of the grid is $7 \times 8 = 56$. Every tested input condition is replicated 6 times. For every tested input condition, the leaf area and tipburn area after harvest are observed, both measured in the unit of mm$^2$. The resulting observations are

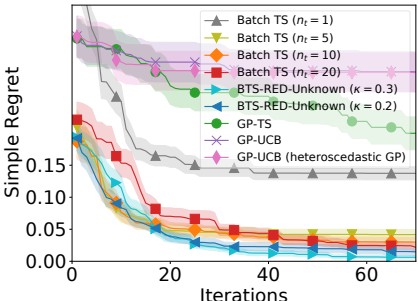

Figure 5: Results for the mean optimization problem for the synthetic experiment (Sec. 5.1) after adding comparisons with some sequential BO algorithms. Sequential BO algorithms are unable to perform competitively with other algorithms which leverage batch evaluations.

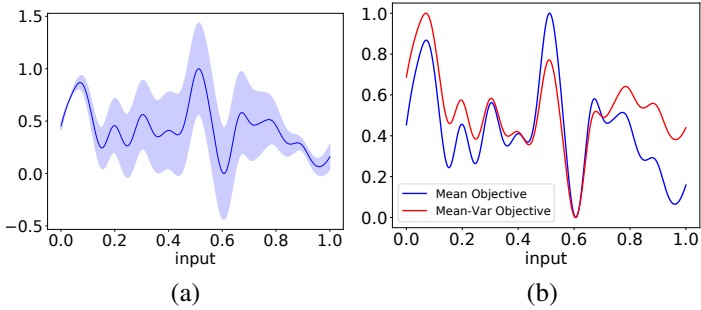

(a)          (b)

Figure 6: (a) The synthetic function used in the experiments for mean-variance optimization in Sec. 5.1. (b) The mean and mean-variance objective functions ($\omega = 0.3$).

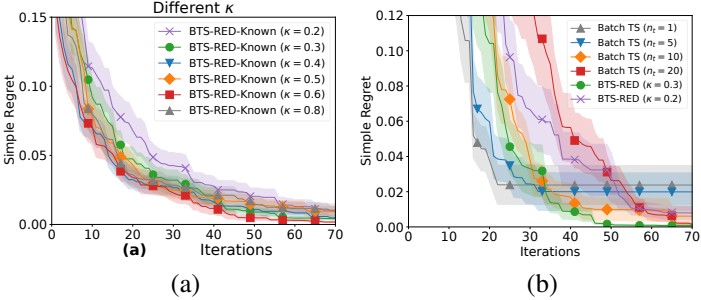

(a)          (b)

Figure 7: (a) Results using more values of $\kappa$ for BTS-RED-Known in the synthetic experiment. (b) Results for the synthetic experiment using the LCB as the report metric.

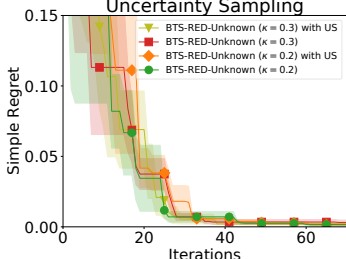

Figure 8: Comparisons of results using uncertainty sampling (US) and random search as the initialization method.

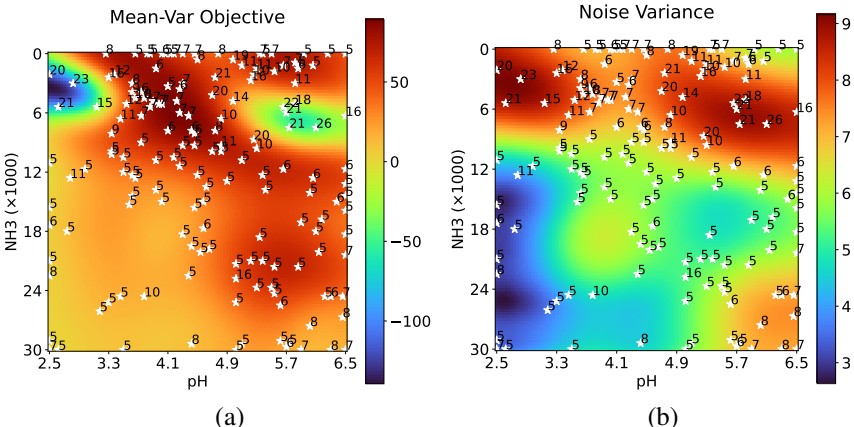

(a)                                                    (b)

Figure 9: Groundtruth (a) mean-variance objective function and (b) noise variance functions for the leaf area, including some selected queries (white stars) and the corresponding $n_t$'s.

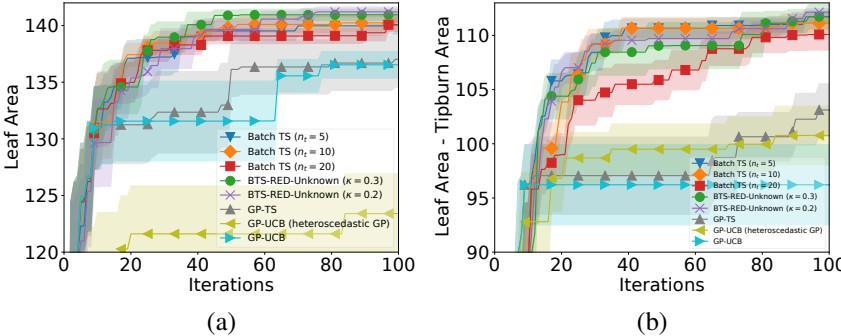

(a)                                                    (b)

Figure 10: Mean optimization for (a) the leaf area and (b) the weighted combination of leaf area and negative tipburn area. The results for some sequential BO algorithms are included here, i.e., GP-TS, GP-UCB and GP-UCB with a heteroscedastic GP. Similar to Fig. 5 for the synthetic experiment, sequential BO algorithms fail to achieve competitive performances with other algorithms which are able to exploit batch evaluations.

used to learn a heteroscedastic GP model from GPglow (`https://gpflow.readthedocs.io/en/develop/notebooks/advanced/heteroskedastic.html`), which is then used to produce the groundtruth mean and variance function used in our experiments.

### H.3 Real-world Experiments on AutoML

In this experiment, we aim to tune two hyperparameters of SVM: the penalty parameter (denoted as $C$) and RBF kernel parameter (referred to as gamma), both within the range of $[0.0001, 2]$. To obtain the groundtrugh mean and variance functions, we firstly construct a uniform 2-D grid of size $80 \times 80$ using the 2-D domain, and then evaluate every input hyperparameter configuration on the grid using 100 different classification tasks (i.e., using the images of hand-written characters from 100 different individuals in the EMNIST dataset). The EMNIST dataset is under the CC0 license. For every tested input hyperparameter configuration on the grid, we record the empirical mean and variance as the groundtruth mean and variance at the corresponding input location. As a result, this completes our construction of the groundtruth mean and variance functions with the input domain being discrete with a size of $80 \times 80 = 6400$.

Figs. 11a and b plot the constructed groundtruth mean and noise variance functions, including the locations of some selected inputs (the selected inputs after every 4 iterations are included) as well as their corresponding number of replications $n_t$'s. Similarly, Figs. 11c and d show the queried inputs and the $n_t$'s of Mean-Var-BTS-RED ($\omega = 0.2$), shown on the heat maps of the mean-variance objective (c) and noise variance functions (d). The figures show that similar to Fig. 3 for the precision agriculture experiment, the majority of our input queries fall into regions with large (either mean or

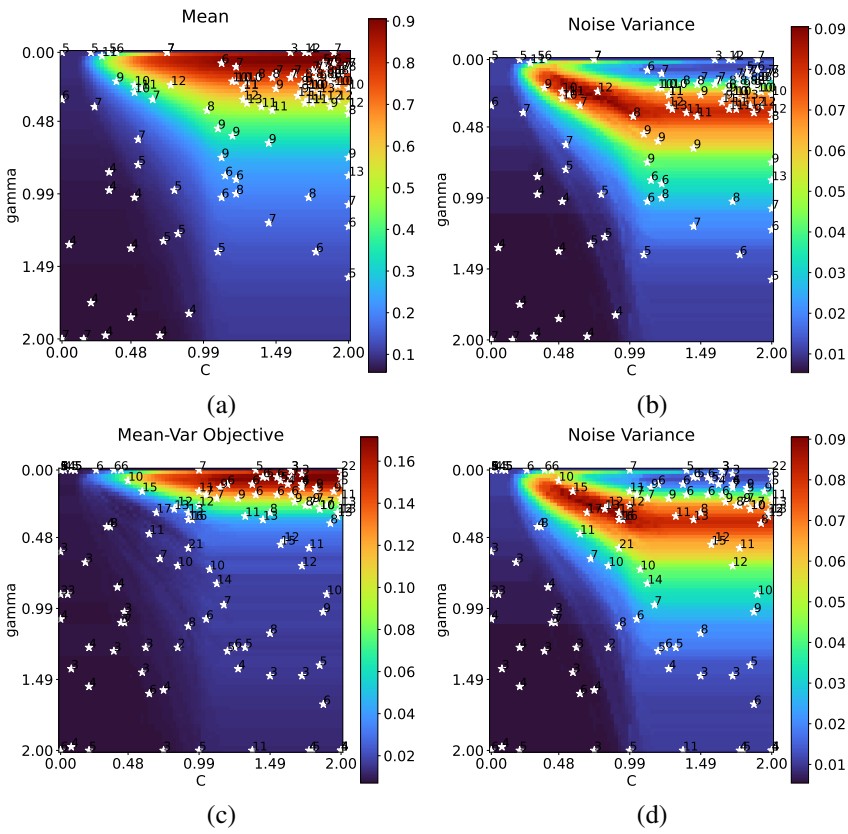

Figure 11: Groundtruth (a) mean and (b) noise variance functions for hyperparameter tuning of SVM, including some selected queries (white stars) and the corresponding $n_t$'s. (c, d): The corresponding plots for mean-variance optimization ($\omega = 0.2$) for hyperparameter tuning of SVM.

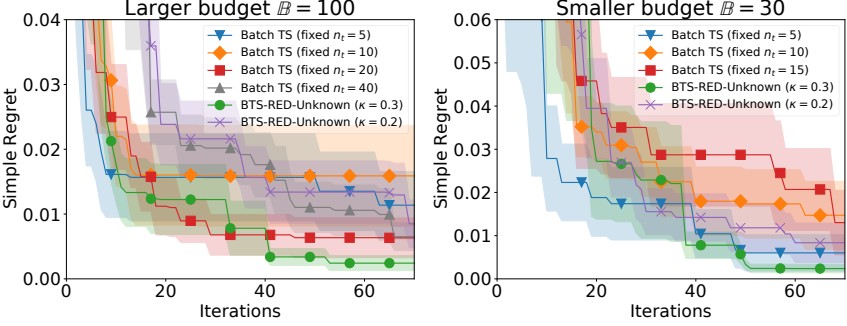

Figure 12: AutoML experiment using different budgets $\mathbb{B} = 100$ and $\mathbb{B} = 30$.

mean-variance) objective function values (Figs. 11a and c) and that the selected number of replications $n_t$ is generally larger at those input locations with larger noise variance (Figs. 11b and d).

## H.4 Additional Experiments with Higher-Dimensional Inputs

The Lunar-Lander experiment requires tuning $d = 12$ parameters of a heuristic controller used to control the Lunar-Lander environment from OpenAI Gym [4]. The controller can be found at https://github.com/openai/gym/blob/8a96440084a6b9be66b2216b984a1c170e4a061c/gym/envs/box2d/lunar_lander.py#L447. The robot pushing task was introduced by [45], and we defer a more detailed introduction to the experimental settings to the work of [45]. Both experiments are widely used benchmarks for high-dimensional BO experiments [16, 20]. In both experiments, for

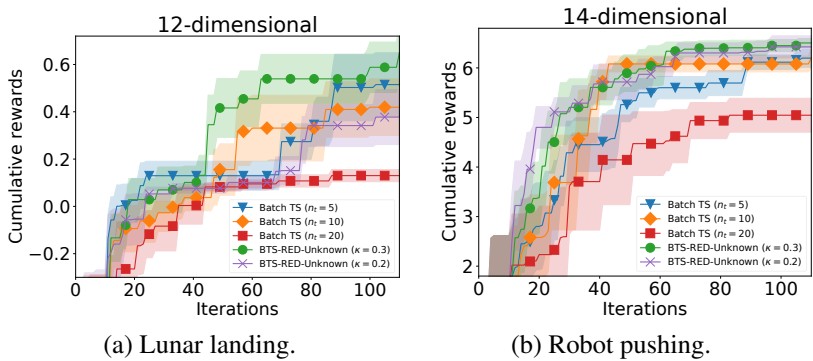

(a) Lunar landing.

(b) Robot pushing.

Figure 13: Experiments with higher-dimensional input spaces.

every evaluated controller parameters (i.e., every evaluated input $x$), the observation (i.e., cumulative rewards) is noisy due to random environmental factors. In addition, the noise may be heteroscedastic. For example, an effective set of parameters which can reliably and consistently control the robot is likely to induce small noise variance, whereas some ineffective sets of parameters may cause radically varying behaviors and hence large noise variances. Therefore, these experiments are also suitable for the application of our algorithms. Since the domain is continuous in these two experiments, here we maximize the acquisition function in every iteration via a combination of random search and L-BFGS-B: in every iteration when we need to maximize the acquisition function, we firstly randomly sample $10,000$ inputs/points in the entire domain and find the point with the largest acquisition function value, and then further refine the search via L-BFGS-B with $100$ random re-starts.

