# OpenReview forum: "Batch Bayesian Optimization For Replicable Experimental Design"
_NeurIPS.cc/2023/Conference — NeurIPS 2023 poster_

### Official Review · Reviewer_VcKK · 2023-06-12

**Soundness:** 2 fair
**Presentation:** 3 good
**Contribution:** 3 good
**Rating:** 6
**Confidence:** 3

**Summary:**

The paper introduces a batch Bayesian optimization method for the setting where experiments can be very noisy, and therefore it is common practice to repeat many experiments. The framework allows for both the selection of the experiment design and how often each experiment is replicated. The authors introduce three method: BTS-RED-Known (for the setting where the noise is known), BTS-RED-Unknown (for the setting where the noise is unknown), and Mean-Var-BTS-RED, for the setting where we want to trade-off finding the optimum while obtaining less noisy observations (called risk-averse optimization).

In every case the experiment designs are chosen using a variation of Thompson Sampling (with scaled variance). The number of replications is then chosen based on the noise level at the experiment design, an upper-bound is used when the noise is unknown. For risk-averse optimization, a linearized sum of the objective and the noise level is used. For every algorithm theoretical bounds are found on the regret, which are proven to be sub-linear. The number of repeated experiments depends on the hyper-parameters $R^2$, but a theoretical justification for its value is given.

Empirically, the method is shown to outperform other Bayesian Optimization algorithms in synthetic examples, and two real world examples: one in Precision Agriculture, and one in AutoML.

**Strengths:**

Originality: To the best of my knowledge, the proposed algorithm is novel. While it combines simple ideas, it does so carefully and effectively. In particular, the theoretical analysis of the algorithms is very strong, providing a choice for the hyper-parameter $R^2$, and proving regret bounds for the algorithm.

Quality: The algorithm is built up from proved and tested methods in the literature, and a strong theoretical justification is given for using them. Empirical evidence of the algorithm performance is given, but in a limited capacity. The creating of a data-set (for the agriculture example) based on real-life experiments makes the experiment particularly strong.

Clarity: The method is well explained, and the theoretical implications are clear. There is a good choice of figures, even if they are difficult to read.

Significance: The paper is well motivated, experiment replication due to noise is common in many areas of science and it is not usually taken into account by classical Bayesian optimization algorithms. Making BO more practical to the wider scientific community is very important, and papers like this one take a big step towards it.

**Weaknesses:**

While the motivation, and justification for the algorithms is clear, I find the empirical evidence to be lacking. My main concerns are:

- The main framework seems to fit a _homoskedastic_ GP to the mean of the replicated data (which will, by design, have very small noise so it should not be a problem). Then, if needed, a second homoskedastic GP is used to model the variance. However, I would argue the most naive and natural solution is to simply use a heteroskedastic GP as the surrogate model in the first place. This is not a problem in itself, however, I believe for a fair comparison against Thompson Sampling a homoskedastic GP should be used. Otherwise the model misspecification could be problematic and lead to poor performance (maybe Figure 1d could be explained by this?). It is unclear from the paper if this is the case. In simpler terms, *it is unclear wether the empirical advantage of BTS-RED comes from a better algorithm, or a better model*. If a heterskedastic GP is used, then is should be clarified, otherwise the paper would benefit greatly from including it.

- The extent of the experiments seems limited. Only a single 1d example is shown, and then both real world examples are 2d. It would be good to include more examples. Similarly, the budget is fixed to $\mathbb{B} = 50$ for all experiments, it would be good to see how the algorithm performs for smaller (and perhaps larger) budgets.

Other minor issues:

- The novelty of Mean-Var-BTS-RED seems overstated, is it simply solving a multi-objective problem through linearization?

- The heuristic for handling unused budget seems a little problematic. Given you get some observations (albeit noisy), if they clearly pointed to a sub-optimal design it seems wasteful to still evaluate the objective further at the specific location.



**Questions:**

- Does *asymptotically no regret*  simply mean that the method converges? Or that convergence is sub-linear?

- Am I correct in assuming that the GPs that you are fitting when using BTS-RED are homoskedastic with a small noise level?

- Is there an intuitive meaning to the variable $R^2$? Or is this a hyper-parameter to choose? I think it would be good to talk about it earlier, as it is often mentioned, but not discussed until page 5.

- When doing Thompson Sampling for for BTS-RED, the variance of the GP is scaled (e.g. line 4 in algorithm 1). Is this done for theoretical guarantees or for some other reason?

- I do not fully understand section 3.2.2 (Upper Bound on Noise Variance Function). If the posterior of $g$ is a GP is the result that $-\mu_{t-1}'(x) + \beta_t' \sigma_{t-1}'(x)$ an upper-bound not obvious? If we talk about the general case where $\epsilon'$ is not necessarily Gaussian, then the posterior is not necessarily a GP which could mean my confusion is just a typo in line 214.

Minor formatting issues:
- I found the figures and their font sizes to be too small and difficult to read.
- In Theorem 3.1 $\tau_{t-1}$ and $\beta_t$ are defined and then not used in the Theorem





**Limitations:**

Limitations are mentioned in the Conclusion section. Other limitations I mentioned (e.g. lacking empirical evaluation) seem like they could be easily addressed by the authors in a single iteration of the paper. There is no obvious negative social impact.

---

> ### Author Rebuttal · Authors · 2023-08-10
>
>
> We thank the reviewer for your constructive feedback.
>
> ---
>
> > - The main framework seems to fit a homoskedastic GP to the mean of the replicated data... (and) a second homoskedastic GP is used to model the variance. However, I would argue the most naive and natural solution is to simply use a heteroskedastic GP as the surrogate model in the first place. This is not a problem in itself, however, I believe for a fair comparison against Thompson Sampling a homoskedastic GP should be used... If a heterskedastic GP is used, then is should be clarified, otherwise the paper would benefit greatly from including it.
>
> You are correct that our algorithms fit two homoskedastic GPs to model the mean and the variance, respectively. If we understand corretly, you are asking whether we have used a homoskedastic or heterskedastic GP for the baseline of batch Thompson sampling (TS) in our experiments: We have used a homoskedastic GP, which is consistent with the homoskedastic GP used in our algorithms and with the previous works on batch TS. As you suggested, we've added an experiment in which we replace the homoskedastic GP in batch TS by a heterskedastic GP. The results (Fig. 4 in our global response above) show that the use of heterskedastic GP leads to comparable (slightly better) performances for batch TS, but it is still consistently outperformed by our algorithms. Therefore, we think this can serve as additional evidence to justify that our empirical advantage comes from better algorithms.
>
> In addition, to the best of our knowledge, using a single heteroskedastic GP as the surrogate model makes it difficult to perform theoretical analyses. In contrast, the use of two homoskedastic GPs in our methods has allowed us to derive strong theoretical guarantees.
>
> ---
>
> > - The extent of the experiments seems limited. Only a single 1d example is shown, and then both real world examples are 2d. It would be good to include more examples.
> Similarly, the budget is fixed to $\mathbb{B}=50$ for all experiments, it would be good to see how the algorithm performs for smaller (and perhaps larger) budgets.
>
> As you suggested, we've added more experiments (see our global response above).
> Firstly, we've added two experiments with **higher-dimensional continuous input spaces** (12d and 14d, see Fig. 1 in our global response). Our methods still consistently achieve compelling performances (especially with $\kappa=0.3$, which is consistent with our original experiments, see lines 258-261).
> We've also added two experiments with different values of the budget $\mathbb{B}$: $\mathbb{B}=100$ and $\mathbb{B}=30$ (see Fig. 2 in our global response), in which the performance advantages of our methods (especially $\kappa=0.3$) are consistent with those shown in our original paper (with $\mathbb{B}=50$).
>
> ---
>
> > - The novelty of Mean-Var-BTS-RED seems overstated, is it simply solving a multi-objective problem through linearization?
>
> We have indeed designed our Mean-Var-BTS-RED algorithm based on the scalarization/linearization technique, which is a common technique for multi-objective optimization. However, our theoretical analysis for Mean-Var-BTS-RED are novel to the best of our knowledge, and they posed additional technical challenges compared with the analysis of our BTS-RED-Unknown.
>
> ---
>
> > - The heuristic for handling unused budget seems a little problematic. Given you get some observations (albeit noisy), if they clearly pointed to a sub-optimal design it seems wasteful to still evaluate the objective further at the specific location.
>
> What you have suggested here is indeed an interesting potential extension. However, as you have also alluded to, the observations are likely very noisy due to the incomplete evaluations, and hence using these potentially noisy observations to decide whether to continue the unfinished experiment may lead to unreliable decisions. Therefore, this extension is likely to require new algorithmic designs, which we will explore in future works. Thank you for the suggestion.
>
> ---
>
> > - Does asymptotically no regret simply mean that the method converges? Or that convergence is sub-linear?
>
> To clarify, our methods being asymptotically no-regret means that our cumulative regret is sub-linear. This also implies that our simple regret asymptotically goes to 0; intuitively, this means that our methods are guaranteed to be able to query the global optimum asymptotically, in other words, our method is guaranteed to converge. Please see lines 90-104 for more details.
>
> ---
>
> > - Is there an intuitive meaning to the variable $R^2$? Or is this a hyper-parameter to choose? I think it would be good to talk about it earlier, as it is often mentioned, but not discussed until page 5.
>
> The parameter $R^2$ can be seen as the *effective noise variance* for every observations (lines 122-124). We have used our theoretical regret bound to derive a guideline on how to choose $R^2$ (lines 171-181), which we have indeed followed in our experiments (lines 249-256). To clarify, we had in fact already discussed about $R^2$ on page 3 (lines 118-124), and we'll revise this part to make the discussion clearer.
>
> ---
>
> > - When doing Thompson Sampling for for BTS-RED, the variance of the GP is scaled. Is this done for theoretical guarantees?
>
> Yes, this is done for the theoretical guarantees following the previous works [6].
>
> ---
>
> > - I do not fully understand section 3.2.2 (Upper Bound on Noise Variance Function). If the posterior of $g$ is a GP, is the result that $-\mu'\_{t-1}(x)+\beta'\_t\sigma'_{t-1}(x)$ is an upper-bound not obvious?
>
> This upper bound naturally arises only given the assumption that the noise $\epsilon'$ is sub-Gaussian. We think that it may not be obvious to see that this assumption is reasonable, and hence we have justified why this assumption is reasonable in lines 201-210.
>
> ---
>
> Thank you again for your insightful comments. We hope our additional clarifications and results could improve your opinion of our paper.

---

> > ### Comment · Reviewer_VcKK · 2023-08-10
> > **Response to Rebuttal**
> >
> > Thank you for the very detailed response and for the additional experiments. I think most of my concerns have been well addressed. It is encouraging to see that the method holds up against heteroskedastic TS, I would still prefer if this was compared against in all benchmarks as I think it is the more natural baseline, but I guess it is not strictly necessary. I seem unable to edit my review at the moment, but I will upgrade my score when I get the chance.

---

> > > ### Author Response · Authors · 2023-08-12
> > > **Thank You for You Comments**
> > >
> > > We're happy to hear that most of your concerns have been well address, and we highly appreciate for your improved evaluation of our paper.
> > >
> > > We'll also revise the paper following your comments, which we believe will significantly improve our work.

---

### Official Review · Reviewer_2mxf · 2023-07-01

**Soundness:** 3 good
**Presentation:** 3 good
**Contribution:** 3 good
**Rating:** 6
**Confidence:** 4

**Summary:**

This paper introduces an algorithm for selecting both sampling locations and the number of replications in the context of heteroskedastic Bayesian Optimization. The authors propose to use Thompson sampling for batch candidate selection and a scheme for determining the number of replications for each element of the batch based on an effective noise variance. The authors propose three different variants of their algorithm, covering both the noiseless and noisy cases, as well as the mean-variance optimization case. They prove asymptotic regret bounds for each case (based on which they suggest the choice of effective noise variance). Finally, the paper empirically compares the proposed algorithm against baselines on synthetic and real-world problems.

**Strengths:**

- This is a well-motivated problem and of interest to the Bayesian Optimization research community as well as to practitioners.
- While the basic ideas aren't novel, the specific approach taken is. It is also quite simple and easy to implement.
- The theoretical results appear sound and are useful, especially the guidelines for how to choose the effective noise variance. The technical contributions are solid and non-trivial.
- The paper is generally well written and the contributions are stated clearly.



**Weaknesses:**

- Theoretical results:
    - While the results are interesting, it's not really clear to me to what extent the asymptotic rates from this approach differ from those that would be achieved by uniform sample allocation (see my questions below). This seems like a missing aspect to me that the authors should address.
- Empirical results:
    - It feels like the authors could have attempted harder to compare against other baselines.
        - The comparison is only against sequential BO algorithms, which have a clear disadvantage in that they cannot, well, use batch evaluations (this includes RAHBO). While the authors acknowledge that, the obvious thing to do here would be to compare against parallel (batch) BO algorithms (using uniform budget allocation) such as qEI, qUCB, etc. that are readily available in the literature (and are implemented in many BO libraries). This would help better understand the performance of the proposed method in absolute terms (rather than solely in terms of improving over batch TS due to non-uniform allocation.
        - Similarly, even though [34] may be heuristic, it would still be useful to compare against empirically (if that is reasonably straightforward to do).
    - The improvements relative to a basic baseline such as simple batch TS do not strike me as particularly impressive.
        - This may be due to the simple problems that were used. It would be helpful to see how the algorithm performs on higher dimensional problems (max dimension is 2) and what effect the dimension has on the significance of the ability to be able to allocate budget in a nonuniform fashion.
        - Maybe these results are more impressive than I think, but if that's the case maybe the authors can try to explain this better?
- Figures: The figures in the MT are tiny and extremely hard to read. I suggest moving some of the technical details from section 3 to the appendix and generally tighten up the writing and make some readable figures instead.

**Questions:**

- Asymptotic no-regret:
    - Wouldn't a uniform sampling strategy achieve the same? Are there any improved regret bound rates that we can get from the proposed approach vis-a-vis a uniform allocation?
    - *Note*: I am not questioning the practical value of doing non-uniform allocation, but I wonder if the bound is more interesting - especially even with a uniform allocation I would assume that in a finite space you should asymptotically achieve similar rates (log(B)/sqrt(B)) in terms of the budget as if you used the proposed strategy. The non-asymptotic regime seems more interesting). Not that this is still with heteroskedastic noise, so this is somewhat different than comments on the homoskedastic noise case the authors make in l164.
- What if the noise is heteroskedastic in the outcome value (rather than necessarily the input)? Can this be exploited in a different way?
- What would a more principled approach for the n_max heuristic for avoiding samples on undesirable regions look like? Could this be handled in a more principled as part of the point generation?
- For the unknown noise case, the GP modeling the empirical noise variance may produce negative predictions. How are you dealing with this? Have you considered modeling the log-variance with a GP instead (a common approach in other similar work)?
- Finiteness of the domain: The discretization approach ("we assume that the domain X is finite, since extension to compact domains can be easily achieved via suitable discretizations [4].") is fine in theory (for regret bounds etc) but can become problematic in practice, especially in higher dimensions where the number of discrete points to sample needs to grow exponentially with the dimension to get similar coverage of the domain, which renders acquisition function optimization challenging. It seems reasonably straightforward to optimzie the acquisition fucniton using gradient-based optimization, maybe the authors may want to comment on that.
- "it is recommended to make nmin larger in experiments where the overall noise variance is large" <- why is that recommended? It will make it easier to estimate the local noise variance, but it's not necessarily clear that this is going to result in better optimization performance.
- "When the noise variance is unknown [...] we approximate sigma_max^2 by the maximum observed empirical noise variance and update our approximation after every iteration" <- This makes a lot of sense, but it feels like something is swept a bit under the rug here. Specifically, what does this do to the regret bound, which  depends on sigma_max^2? Is it still valid (just with an unknown factor)? Or is there additional work to show this?
- The agriculture example is a nice real-world application and a good fit for the algorithm, however, I question the number of iterations that are being evaluated here. How long is a growing cycle until the leaf area and tipburn area can be evaluated? If this is on the order of months then 100 growth cycles seem a lot...
- The regularization parameter lambda depends on the overall horizon T (and the appendix states that this is required for the theoretical results to hold) - however, in practice T may not be know if we are interested in anytime performance of the algorithm. How would one deal with this in practice?
- Why are you optimizing the GP hyperparameters only after every 10 iterations and not after every iteration?
- typo l97: "upper on"

**Limitations:**

Yes

---

> ### Author Rebuttal · Authors · 2023-08-10
>
>
> We thank the reviewer for your detailed and insightful comments.
>
> ---
>
> > Theoretical results: comparision with the theoretical regret bound achieved by uniform sample allocation.
>
> For uniform sample allocation (i.e., we replicate every input a constant $n_0\leq\mathbb{B}$ number of times), the effective observation noise would be $(\sigma_{\max} / \sqrt{n_0})$-sub-Gaussian, where $\sigma_{\max}$ is the maximum noise standard deviation. This would result in a regret bound which can be obtained by replacing the term $\sqrt{R^2 / (\mathbb{B} / \lceil \frac{\sigma^2_{\max}}{R^2} \rceil  - 1)}$ in our regret bound (Theorem 3.1) by $\sigma_{\max} / \sqrt{n_0}$. Here we ignore the non-integer conditions (e.g., ceiling operators) for simplicity. As a result, with our optimal choice of $R^2$ (lines 171-181), the above-mentioned term from our Theorem 3.1 can be simplified to $\sigma_{\max}/\sqrt{\mathbb{B}}$ (details omitted here), which is to be compared with the above-mentioned term $\sigma_{\max} / \sqrt{n_0}$ from uniform sample allocation. Because $n_0 \leq \mathbb{B}$, **our regret bound** (with the scaling of $\sigma_{\max} / \sqrt{\mathbb{B}}$) **is guaranteed to be no worse than the regret of uniform sample allocation** (with the scaling of the $\sigma_{\max} / \sqrt{n_0}$).
>
> Thank you for pointing out this interesting comparison, we'll add the discussions here to the paper after revision.
>
> ---
>
> > Empirical results:
> - Performance on more complex, higher-dimensional problems
> - Comparison against more baselines on batch BO
>
> **We've added two more complex real-world experiments with higher-dimensional input spaces** (Fig. 1 in our global response above), with input dimensions of $d=12$ and $d=14$. Our methods still consistently achieve compelling performances (especially with $\kappa=0.3$, which is consistent with our original experiments, see lines 258-261).
>
> We've compared with batch TS as the representative baseline batch BO algorithm because it's the most natural competitor to our algorithm, and batch TS is both simple and has been found to yield competitive empirical performances (e.g., the TuRBO algorithm [13], which showed impressive performances in various applications, also used batch TS for batch selection). As you suggested, we'll explore comparisons with additional baseline batch BO methods in future works.
>
> ---
>
> > What if the noise is heteroskedastic in the outcome value (rather than necessarily the input)?
>
> To clarify, in our setting, the noise variance $\sigma^2(x)$ varies with the input $x$, and the realized noise $\epsilon\sim\mathcal{N}(0,\sigma^2(x))$ is added to the function value $f(x)$ to produce the outcome $y=f(x)+\epsilon$.
>
> ---
>
> > What would a more principled approach for the $n_{\max}$ heuristic for avoiding samples on undesirable regions look like?
>
> A more principled approach for avoiding samples in undesirable regions would require taking into account the function value $f(x)$ at different $x$. Since the function is unknown, we may instead use the confidence bounds calculated by the GP which contains the function with high probability. We'll explore this in future works.
>
> ---
>
> > The GP modeling the empirical noise variance may produce negative predictions.
>
> We find that in practice, it's very rare for our **upper bound on the unknown noise variance** (line 218) to be negative. In practice, we clip this upper bound to account for such exceptions. This modelling choice is adopted since it naturally allows us to derive our theoretical guarantees, and it also leads to good empirical performances in our experiments. It's interesting to see if modelling the log-variance could further improve our performance.
>
> ---
>
> > Finiteness of the domain: The discretization approach is fine in theory but in higher dimensions, it renders acquisition function optimization challenging.
>
> It's indeed a generic challenge for BO to optimize the acquisition function in high-dimensional continuous input spaces, for which we can adopt commonly used techniques (e.g., L-BFGS-B) in BO. This is exactly what we've done in our added experiments with continuous high-dimensional input spaces (Fig. 1 in global response above).
>
> ---
>
> > why is it recommended to make $n_{\min}$ larger in experiments where the overall noise variance is large?
>
> When the overall noise variance is high, intuitively, an overall larger number $n_t$ of replications is needed. Hence, we recommend a larger number $n_{\min}$ of minimum replications as a potentially useful heuristic.
>
> ---
>
> > "When the noise variance is unknown [...] we approximate $\sigma^2_{\max}$ by the maximum observed empirical noise variance...". What does this do to the regret bound, which depends on $\sigma_{\max}^2$?
>
> We in fact only need to approximate $\sigma^2_{\max}$ when we **choose the parameter $R^2$** by minimizing our derived regret bound (lines 171-181). So, our regret bounds (e.g., Theorem 3.1), which **hold for all values of $R^2$**, are still valid.
>
> ---
>
> > I question the number of iterations in the agriculture example. How long is a growing cycle?
>
> The growing cycle is around 1-2 weeks. Also note that our algorithms in fact already outperform the other methds before 100 iterations.
>
> ---
>
> > The regularization parameter $\lambda$ depends on the overall horizon T  - however, in practice T may not be known.
>
> It's a common practice in BO to assume that the overall horizon $T$ is known [6,33]. When $T$ is not known, we can use the doubling trick commonly adopted in multi-armed bandits to obtain anytime algorithms, or simply use an estimation of $T$.
>
> ---
>
> > Why are you optimizing the GP hyperparameters only after every 10 iterations?
>
> It's a common practice to update the GP hyperparameters after multiple iterations, mainly to save computational cost.
>
> ---
>
> Thank you again for your valuable feedback. We hope our clarifications and additional results could improve your opinion of our paper.

---

> > ### Comment · Reviewer_VcKK · 2023-08-10
> > **Additional Question**
> >
> > Just a follow-up question regarding batching algorithms, stemming from this review. It is my experience that batching methods such as q-EI, q-UCB or Local Penalization can become computationally expensive and scale badly for large batches. Because of this, and looking at the large batch-sizes in the experiments ($ B = 30, 50, 100 $) I thought the choice of Thompson Sampling to be very natural, however, I am wondering how would you expect the algorithm to perform in batch sizes in the range $ B \in \{5, 6, ..., 20 \}$ where other methods could be more natural competition.

---

> > > ### Author Response · Authors · 2023-08-12
> > > **Response to Additional Question**
> > >
> > > Thank you for pointing out this additional advantage of batch Thompson sampling (TS) in terms of computational costs given our large batch sizes. It is also what we have observed in our experiments, i.e., increasing the batch size for batch TS does not siginificantly increase the computational cost. We'll add it to the paper as an additional justification for our choice of TS for batch selection.
> > >
> > > When $\mathbb{B}$ gets smaller, we expect the behavior of our algorithm to become more and more similar to standard batch TS (without replications). To see this, note that as $\mathbb{B}$ becomes smaller, our choice of $R^2$: $R^2=\sigma^2_{\max}(\sqrt{\mathbb{B}}+1)/(\mathbb{B}-1)$ (line 175) will become larger; as a result, our selected number of replications $n\_t^{(b)}=\lceil \sigma^2(x^{(b)}\_t) / R^2 \rceil$ (line 5 of Algo. 1) will become smaller. In the extreme case where $\mathbb{B}=4$, we have that $R^2=\sigma^2_{\max}$ and consequently $n_t^{(b)}=\lceil \sigma^2(x^{(b)}_t) / R^2 \rceil=1$, which means our algorithm reduces to standard batch TS (without replications).
> > > Therefore, when $\mathbb{B}$ is small, we expect our algorithms to behave similarly to standard batch TS. Given the good empirical performances of batch TS compared with other batch BO algorithms shown in the literature [13,20], we expect our methods to perform competitively as well.
> > >
> > > To summarize, we think our algorithms perform on par with standard batch TS when $\mathbb{B}$ is small ($\mathbb{B}=4-20$) and consistently outperforms batch TS (as shown in our experiments) when $\mathbb{B}$ is large ($\mathbb{B}>30$). This is in fact well aligned with the motivation of our work, i.e., in precision agriculture, $\mathbb{B}$ usually takes values within the range $\mathbb{B}=50-100$ according to the plant biologists we are collaborating with. We'll also clarify this after revision. Thanks for pointing this out.

---

> > ### Comment · Reviewer_2mxf · 2023-08-12
> >
> > Thanks for the detailed response. A number of my concerns were addressed, thanks also for adding the higher-dimensional comparisons.
> >
> > However, I still feel like I don't really have a sense of how well the proposed approach works compared to other non-TS baselines. While reviewer VcKK points out that TS is well suited to large batch sizes, $\mathbb{B}$ is the total replicate size - with $n_t$ set to a minimum of 5, even with $\mathbb{B}$ the effective batch size is (at most) 10, which is definitely well within the realm of other batch acquisition functions such as qUCB or q(N)EI or other even less computationally demanding heuristics such as the penalization approach of Gonzalez et al. (2016).
> >
> > J. Gonzalez, Z. Dai, P. Hennig, and N. Lawrence. Batch bayesian optimization via local penalization. In A. Gretton and C. C. Robert, editors, Proceedings of the 19th International Conference on Artificial Intelligence and Statistics, volume 51 of Proceedings of Machine Learning Research, pages 648–657, Cadiz, Spain, 09–11 May 2016. PMLR.

---

> > > ### Comment · Reviewer_VcKK · 2023-08-13
> > >
> > > This is a good point. I do think the choice of TS is more natural, but I agree comparing to baselines using qEI or Local Penalization with $n_t \geq 5 $ would make the paper stronger. Also, it could even boost the performance of BTS-RED as the experiment selection could be done using these other baselines (e.g. introduce BqEI-RED or BqUCB-RED etc...) albeit the theoretical guarantees would be lost without further work.

---

> > > > ### Author Response · Authors · 2023-08-15
> > > > **Thank You for Your Comment**
> > > >
> > > > We agree that it is an interesting future direction to incorporate our method (e.g., our adaptive number of replications depending on the noise variance) into other batch BO algorithms (such as qUCB, q(N)EI and the penalization approach of Gonzalez et al. (2016)) to explore whether we can achieve similar or better empirical performances as our BTS-RED.
> > > >
> > > > Thank you for the suggestion and we will add this to the paper as a future topic.

---

> > > ### Author Response · Authors · 2023-08-15
> > > **Thank You for Your Comment**
> > >
> > > We agree that incorporating replications (with a fixed number $n_t$ of replications) into other batch BO algorithms (such as qUCB, q(N)EI and the penalization approach of Gonzalez et al. (2016)) can reduce their effective batch size and hence their computational cost, and it is interesting to explore comparisons of our methods with these methods in future work. In addition, as Reviewer VcKK has commented, it is also an interesting future topic to incorporate our method (e.g., our adaptive number of replications depending on the noise variance) into these other batch BO algorithms.
> > >
> > > Thank you and we will add the discussions here to the revised paper.

---

### Official Review · Reviewer_4YBP · 2023-07-09

**Soundness:** 3 good
**Presentation:** 2 fair
**Contribution:** 2 fair
**Rating:** 4
**Confidence:** 3

**Summary:**

The paper proposes a batched Bayesian optimization with heterodescadic noise and Thompson sampling. Further, it proposes an extension where not only function is minimized but also a robust variant objective which also incorporates observational noise variance. It provides some regret analysis relying on prior works and experimental comparison on real life benchmarks from agriculture.


**Strengths:**

Originality
---------
The use of Thompson sampling with heteroscedastic noise is novel.

Significance
-----------
1. Authors test their algorithm in real-world settings.
2. I think the pattern of trying to repeat measurements is indeed omnipresent in the life sciences, and methodologies which automatically incorporate it are of benefit to practitioners.

Clarity & Quality
--------
I was able to understand the paper, however the exposition is very dense.

**Weaknesses:**

The proof techniques used here are classical used in prior works mainly Chowbury & Gopalan 2017 and Desaultes et. al. 2014. Namely the RKHS proof of Gopalan and Batched version of it using the uncertainty sampling there. This work provides a Thompson sampling look at heteroscedastic bandits but the theoretical results are only of marginal interest to the community since they are straightforward extension of prior techniques.

The focus on Thompson sampling seems arbitrary. There are works which can address heteroscedastic as well as robust optimization but do not use Thompson sampling such as Kirschner & Krause 2018 and Makarova e.t al. 2021. Why Thompson sampling in particular is interesting remains unanswered.

The fact one performs repetitions is not theoretically motivated, but instead only practically motivated; or at least I took the liberty to make this conclusion. Information theoretically speaking, repeating the measurements might not be the best thing to do to decrease uncertainty overall. In fact probably optimizing an allocation over all points can lead to a better solution in long run (over multiple iterations). However, I think studying the problem where the repetitions are a constraint on the measurement scheme is much better motivation for this work, since life-sciences do have certain setup-costs per experiment.

**Questions:**

1. Why do you put heavy focus on the theory? Are the theoretical bounds important for your real-life experiments?
2. You provide a outline how to choose R, but is this what you use in your real lab-experiments.
3. In real experiment - do you use uncertainty sampling as in the theorems?


I my opinion this provides limited novelty towards broader NeurIPS crowd. I think as a case study of Bayesian optimization with heteroscedasdic noise its very nice, and a paper more focused towards actual challenges with precision agriculture would be much more fitting. I think studying the problem where the repetitions are a constraint on the measurement scheme is much better motivation for this work.

**Limitations:**

Authors adequately addressed the limitations and, if applicable, potential negative societal impact of their work.

---

> ### Author Rebuttal · Authors · 2023-08-10
>
>
> We thank the reviewer for your insightful comments.
>
> ---
>
> > The proof techniques used here are classical used in prior works... but the theoretical results are only of marginal interest to the community since they are straightforward extension of prior techniques.
>
> Although we have used some proof techniques from the previous works, we think that our analyses are in fact not straightforward extension of prior techniques. **Rigorously and coherently integrating the different components into our overall analytical framework required careful treatment and non-trivial efforts**. Therefore, we think that our technical theoretical contributions are in fact "solid, non-trivial" and "strong" (as commented by Reviewer 2mxf and Reviewer VcKK).
>
> Moreover, our algorithmic deisgns (e.g., our adaptive selection of the number of replications to guarantee a small effective noise level, lines 116-125), which are indispensable for our theoretical analyses to be hold, are also novel and can be of broader interest to the community. More importantly, in addition to our contributions from the pure theoretical perspective, **our theoretical results have also provided useful practical guidelines for our empirical implementation**. Specifically, our regret upper bound has provided us a natural and principled way to set the effective noise variance $R^2$ (lines 171-181), which we have indeed followed in our experiments (lines 249-256) and led to compelling empirical performances.
>
> In addition, we have also performed empirical evaluations in real-world experiments for precision agriculture (using real-world data from precision agriculture) and AutoML, which we think constitute important empirical contributions and hence demonstrate the potential of our proposed methods in solving real-world problems.
>
> ---
>
> > The focus on Thompson sampling seems arbitrary... Why Thompson sampling in particular is interesting remains unanswered.
>
> Our focus on Thompson sampling (TS) is in fact not arbitrary. Instead, we have adopted TS because its inherent randomness makes it particularly suitable for selecting a batch of inputs [20] in the batch setting we focus on (lines 44-45). In fact, using the inherent randomness of TS for batch selection is a simple and well-established method in BO (e.g., see [13, 20]), which both allows for the derivation of theoretical guarantees [20] and leads to strong empirical performances (e.g., the TuRBO algorithm from [13]). Thank you for pointing this out, and we'll clarify this after revison to avoid confusion.
>
> ---
>
> > The fact one performs repetitions is not theoretically motivated, but instead only practically motivated... I think studying the problem where the repetitions are a constraint on the measurement scheme is much better motivation for this work, since life-sciences do have certain setup-costs per experiment.
>
> You are correct that performing replications is mostly motivated from practical applications. Specifically, it is important to explicitly replicate each condition because in problems with large and heterogeneous observation noises, (1) replicating each input condition leads to more reliable observations and has indeed been repeatedly found [2,25,34] to improve the performances in such problems (lines 23-26), and (2) replicating each conditon is indeed what practitioners do in these real-world problems such as precision agriculture (according to the plant biologists we are collaborating with).
>
> We agree that it will further strengthen our motivation by considering more real-world scenarios with motivations or requirements to perform replications, e.g., when performing new experiments with a different input condition requires expensive experimental setups as you suggested. We'll follow your suggestion and will revise the paper to discuss them as additional motivations. Furthermore, it is also an interesting topic to theoretically show the advantage of performing replications, which we'll also explore in future works.
>
> Thank you very much for the suggestion.
>
> ---
>
> > 1. Why do you put heavy focus on the theory? Are the theoretical bounds important for your real-life experiments?
>
> > 2. You provide a outline how to choose R, but is this what you use in your real lab-experiments.
>
> Firstly, our theoretical bounds can serve as assurance for practioners deploying our algorithms. For example, similar to many previous works on BO with theoretical guarantees, **the fact that our algorithms are asymptotically no-regret can serve as a hallmark indicating that our algorithms are well-behaved**.
> Secondly, our theoretical results have indeed provided us useful and practical guidelines on how to set the effective noise variance $R^2$, which is the most important parameter in our algorithm (lines 171-181). More importantly, **we have indeed followed this theoretical guideline to choose $R^2$ in our real-world experiments** (lines 249-256).
>
> ---
>
> > 3. In real experiment - do you use uncertainty sampling as in the theorems?
>
> In our experiments, we have used the simpler random search instead of uncertainty sampling as the initialization method. This is because it has been reported in previous works that uncertainty sampling is usually only a theoretical requirement and other initialization methods often lead to similar performances in practice [20]. To corroborate this, we have added an experiment using uncertainty sampling as initialization (Fig. 3 in the global response above), and the results show that it indeed leads to very similar empirical performances.
>
> ---
>
> Thank you again for your valuable comments. We hope our clarifications and additional results could improve your opinion of our paper.

---

> > ### Comment · Reviewer_4YBP · 2023-08-14
> >
> > Thank you for your responses. I still believe the heavy focus on theory is to the detriment of the exposition of work.
> >
> > I also noticed comment of the above reviewer who spotted a modelling issue by modelling $\sigma^2$. This is yet again a decision where authors chose a decision which is easier to analyze with current analysis at the expanse of practice.
> >
> > Re: your choice of Thompson sampling. There are many paper on batch-BO where they motivation is that Thompson sampling does not lead to sufficient diversity of batches. So then, what is the conclusion. But I perfectly accept your choice for Thompson sampling. This was not criticism of the choice. I think its fine to make choices, I was just wondering if there is a good reason for it.
> >
> > I find your answer to the optimal choice of R^2 according to the theory dishonest. You do say you use the theory, but at the same time in the next line you say its too conservative and introduce a kappa which you tune to get good results. I mean, you use a scalar variable to tune another scalar variable, so what is the point here? So what if I used kappa=0.1, would it work even better? How universal is this scaling? This is a universal scaling for the whole experiment depending on the budget, which is possible to tune in simulation.
> >
> > My main concern up to this point is with the presentation, I feel that what has been said in the rebuttal is way more interesting that the actual contents of the paper.

---

> > > ### Author Response · Authors · 2023-08-15
> > > **Thank you for further comments**
> > >
> > >
> > > Thank you for further comments.
> > >
> > > ---
> > >
> > > > I still believe the heavy focus on theory is to the detriment of the exposition of work.
> > >
> > > We believe that theory is as important as practice, and **both our theoretical and empirical contributions are integral and indispensable components of our paper**. Although we agree that it is important to design an algorithm which works well empirically, however, **it is of equal importance to provide a theoretical guarantee to an algorithm** which can serve as an assurance for practioners deploying this algorithm and can guarantee the correct behavior of the algorithm. Our theory is also important for the exposition of our work, because it allows our presentation to be more rigorous, principled, and non-ambiguous.
> > >
> > > ---
> > >
> > > > I find your answer to the optimal choice of R^2 according to the theory dishonest. You do say you use the theory, but at the same time in the next line you say its too conservative and introduce a kappa which you tune to get good results. I mean, you use a scalar variable to tune another scalar variable, so what is the point here? So what if I used kappa=0.1, would it work even better? How universal is this scaling? This is a universal scaling for the whole experiment depending on the budget, which is possible to tune in simulation.
> > >
> > > We respectfully disagree on this. Here we are using the theoretical value for $R^2=\sigma^2_{\max}(\sqrt{\mathbb{B}}+1) / (\mathbb{B}-1)$ as a general guideline for our practical implementation. More specifically, we have followed the dependency (of this theoretical value) on $\sigma^2_{\max}$ and $\mathbb{B}$, and introduced an additional multiplies $\kappa$ to account for the potential conservativeness of the theoretical analysis. If we ignore this theoretical value and instead directly tune $R^2$ as a scalar, we wouldn't be able to account for the theoretically inspired dependency of $R^2$ on $\sigma^2_{\max}$ and $\mathbb{B}$. As a result, our algorithm wouldn't be able to automatically adapt to different values of $\sigma^2_{\max}$ and $\mathbb{B}$ in a principled way. Therefore, our design ensures that our choice of $R^2$ is universal in the sense that a single value of $\kappa$ (i.e., $\kappa=0.3$) allows us to achieve good empirical performances in most of our experiments (which is indeed what we have shown). On the other hand, if we instead directly tune $R^2$ as a scalar, we cannot find a universal way to set $R^2$ because it cannot adapt to the different values of $\sigma^2_{\max}$ and $\mathbb{B}$.
> > >
> > > In fact, **it is  a common practice** in Bayesian optimization to use the theoretical value as a general guideline (rather than following the exact theoretical value) for the practical implementation. For example, when choosing the $\beta_t$ parameter in the GP-UCB algorithm [33] (which is used to tune the weight between the GP posterior mean and standard deviation), a common practice is also to only use its theoretical value as a general guildine and simply apply some fixed multiplier. For example, the representative work of [a] below has set $\beta_t=0.2d\log(2t)$ (see Section 4.4 of [a]).
> > >
> > > [a] High Dimensional Bayesian Optimisation and Bandits via Additive Models
> > >
> > > ---
> > >
> > > > My main concern up to this point is with the presentation, I feel that what has been said in the rebuttal is way more interesting that the actual contents of the paper.
> > >
> > > We will follow you suggestion to try to make the paper easier to read by giving more intuitions rather than detailed derivations/proofs, and we will also add what we included in the rebuttal to the paper after revision.

---

### Official Review · Reviewer_QJyu · 2023-07-10

**Soundness:** 3 good
**Presentation:** 3 good
**Contribution:** 2 fair
**Rating:** 5
**Confidence:** 4

**Summary:**

The authors propose three methods for Bayesian Optimization under the constraint of batch sampling and with significant heteroscedastic aleatoric uncertainty assumed. Their methods BTS-RED-Known assumes knowledge of the variance function, BTS-RED-Unknown does not assume the variance function is known and fits a GP to the negative variance function and Mean-Var-BTS-RED, also assumes no knowledge of the variance function but uses the learned variance function to enable risk averse BO by optimizing for a weighted combination of the mean function and the negative variance.

The authors prove their methods are asymptotically no-regret. The methods are tested on synthetic as well as real-world (precision agriculture and hyperparameter optimization) setups.

**Strengths:**

- The paper is well written and clear.
- The BTS-RED methods are simple and intuitive, it would be easy to implement and apply them to new problems.
- The method performs well on the limited (see below) empirical evaluation conducted.
- The authors prove the BTS-RED methods are asymptotically no-regret.
- The Mean-Var extension for risk averse BO is interesting.

**Weaknesses:**

- The method is developed for discrete valued domains, IMO extension to real-valued domains may not be trivial (see limitations section).
- The experiments are all conducted on very low dimensional, discrete domains with small bounds. I would have liked to see evaluations on real-valued domains and higher dimensional problems e.g. hyperparameter tuning with 10-20 mixed type hyperparameters. Currently the empirical evaluation is quite limited as a result.
- For the synthetic and precision agriculture experiments, the authors guarantee the ground truth function is of the heteroscedastic GP model class that their method assumes. This is not unreasonable, but given that there is only one further experiment in the paper which comes from a real-world unknown function (hyperparameter tuning experiment) this limits the robustness of the empirical evaluation as it gives the BTS-RED methods quite an advantage over competitor methods which do not have a heteroscedastic GP function representation. I would have liked to see further evaluated on more unknown ground truth functions.
- Overall I am slightly concerned with the substantiveness of the contribution of the paper. The use of a heteroscedastic GP and Thompson sampling for BO is well established. The extension to batch sampling is interesting, but to what extent the BTS-RED methods interestingly integrate batch structure seems limited to me (see questions below). The Mean-Var extension is again interesting but fairly straightforward. All together the novelty and contribution seems somewhat limited compared to prior work e.g. [16].

**Questions:**

- I was surprised to see that the second GP was used to model $- \simga^2$. Is this correct? Why is the GP not used to model $log \simga^2$ as is more standard in the literature as it can take on values on the full real-valued range?
- I am not sure I fully understand the implementation of non-batch BO algorithms in the experimental section. It would seem to me that a non-batch algorithm should have an advantage in that they should sample 1 point, make an observation, update the function representation and sample again. Of course this may make them infeasible to apply to problems with batch structure e.g. precision agriculture. However in the experiments these methods are performing very poorly, so I am assuming they are not implemented as above. Are you using the non-batch methods in a batch-wise fashion? I assume this would mean that all these methods would in effect simply sample the exact same point many times per batch until the budget is used up? If my understanding is correct, I am not sure this is an interesting comparison and it would be interesting to evaluate these methods as they were intended to be used as a upper bound on performance compared to methods with the batch constraint.
- Why is there a need to compute the empricial mean over the replications (line 9, algorithm 1)? It would seem to me that adding (x, y_1), ..., (x, y_n) data points to the training data would be a better use of the data as it would explicitly weight observations with more replications more highly. It seems to me that you are losing the weighting by the number of replications be computing the empirical mean of the observations and treating all such aggregated data points as equal (line 10)?
- Do I understand correctly that in the various BTS-RED methods, the batch structure is only accounted for in the heuristic regarding replicating the same point too many times? Otherwise sample points are selected independently, with no cross-datapoint correlations accounted for and no knowledge of the prior points in batch? This would seem like a significant weakness of the method, and it seems feasible to modify the method to account for this, see for example "BatchBALD: Efficient and Diverse Batch Acquisition for Deep Bayesian Active Learning" for a similar treatment in the related active learning space.

**Limitations:**

The authors address some limitations of the method e.g. the use of a heuristic to handle unused budget in the last iteration of the algorithm. However I would liked to have seen a more comprehensive discussion of other limitations:

- The method is tested on discrete valued domains, it is claimed extension to the real-valued domain is feasible, however it is not clear to me that the heuristics introduced to ensure that the same point is not sampled repeatedly can be trivially extended to a real-valued domain where it would be possible to sample a very near by (essentially identical) point without adding to the count of the number of replications for the originally sampled point.
- Heuristics are introduced to ensure the same point is not sampled too many times and the method introduces new hyperparameters which cannot all be set following apriori guidelines.

---

> ### Author Rebuttal · Authors · 2023-08-10
>
>
> We thank the reviewer for your valuable feedback.
>
> ---
>
> **Clarification on Batch Selection and the Heuristic:**
>
> > ...the batch structure is only accounted for in the heuristic regarding replicating the same point too many times...
>
> > ...it is not clear to me that the heuristics introduced to ensure that the same point is not sampled repeatedly can be trivially extended to a real-valued domain...
>
> > ...to what extent the BTS-RED methods interestingly integrate batch structure...
>
> To clarify, **our methods do not need additional heuristics to ensure that the same point is not sampled repeatedly**. Instead, the diversity of the selected points in a batch is achieved by **the inherent randomness of the Thompson sampling** (TS) strategy (lines 44-45). In fact, using the inherent randomness of TS for batch selection is a simple and well-established method in BO (e.g., see [13, 20]), which both allows for the derivation of theoretical guarantees [20] and leads to strong empirical performances (e.g., the TuRBO algorithm from [13]).
> More importantly, we think that the strong theoretical guarantees of our algorithms (e.g., our algorithms are asymptotically no-regret, e.g., see line 157) indicate that our batch selection strategy is principled rather than heuristic.
>
> Regarding the heuristic you mentioned, if we understand correctly, we think you are referring to the heuristic of a maximum number of replications mentioned in lines 129-130. As a clarification, this heuristic is only needed **to further improve our empirical performance**, and is hence not an essential part of our algorithmic design. As a result, this heuristic **does not affect the ability of our algorithm to solve problems with continuous real-valued domains** (please see our next response).
>
> ---
>
> > More real-world experiments with higher-dimensional, continuous domains.
>
> As we've clarified above, applying our methods to continuous real-valued domains is indeed feasible and in fact trivial. **We have added two real-world experiments with high-dimensional continuous input domains** (see Fig. 1 in our global response above), with input dimensions of $d=12$ and $d=14$. Our methods still consistently achieve compelling performances (especially with $\kappa=0.3$, which is consistent with our original experiments, see lines 258-261). These additional results are further demonstrations of the practicality and real-world potentials of our methods, and we'll add them to the paper after revision.
>
> ---
>
> > Substantiveness of Our Contributions
>
> Here we clarify our novelty and contributions. In addition to the novelty of our algorithmic design (e.g., our adaptive selection of the number of replications depending on the noise variance), a major part of our novelty and contributions (compared with the previous works such as [16]) comes from our theoretical analysis. Specifically, we have shown that our algorithms are asymptotically no-regret, and are guaranteed to improve with a larger budget $\mathbb{B}$ or smaller noise level. Moreover, our theoretical results have provided guidelines on the practical implementation of our algorithms (lines 171-181), which we have indeed followed in our experiments. Lastly, we have empirically evaluated our methods in real-world problems with large and heterscedastic noise (e.g., the experiment using real-world data from precision agriculture in Sec. 5.2) and shown that our methods achieve competitive performances.
> Therefore, we think that our algorithmic design, theoretical analysis and empirical experiments constitute important contributions.
>
> ---
>
> > Using the second GP to model $-\sigma^2$ (instead of $\log\sigma^2$).
>
> This is because it allows us to naturally derive our theoretical guarantees. Moreover, this choice has indeed helped our algorithms achieve compelling empirical performances, as demonstrated by our experiments. We'll explore modelling $\log\sigma^2$ in future works to see if it leads to further performance gains.
>
> ---
>
> > Implementation of non-batch BO algorithms
>
> The reason why batch methods are favored in practice (over non-batch methods) is due to their ability to perform paralell evaluations. So, in our experiments, we have adopted a more realistic comparison, which allows the practical advantage of batch methods to be seen more clearly. Specifically, in our figures, every iteration represents 1 batch, and hence non-batch methods can be seen as having a batch size of 1.
>
> ---
>
> > Why is there a need to compute the empricial mean over the replications (line 9, algorithm 1)? ... It seems to me that you are losing the weighting by the number of replications...
>
> The use of the empirical means over replications is theoretically motivated, i.e., it allows all our observations to have the same effetive noise variance $R^2$ and hence serves as the foundation for our theoretical analyses. Moreover, the weighting by the number of replications is in fact implicitly accounted for by the effective noise variance $R^2$. This is because for every queried input, its number of replications is adaptively selected to ensure that the effective noise variance for its observation is upper-bounded by $R^2$ (see lines 116-125 for detailed explanations).
>
> ---
>
> > Our hyperparameters.
>
> Regarding our hyperparameters, we have discussed how our hyperparameters are set (Sec. 5, first paragraph). Most of our hyperparameters are kept at the same values in all our experiments, and regarding our most important hyperparameter $\kappa$, we have shown that the values of $\kappa=0.2,0.3$ (especially $\kappa=0.3$) consistently lead to competitive performances in all our experiments. So, we think our adopted hyperparameter values can serve as good recommendations for the practical deployment of our methods.
>
> ---
>
> Thank you again for your insightful comments. We hope our additional clarifications and experiments could improve your opinion of our paper.

---

> > ### Comment · Reviewer_QJyu · 2023-08-21
> > **Rebuttal response**
> >
> > I thank the authors for their clarifying comments and in particular the new experiments on a real valued domain they have added (they certainly improve the paper), I will update my score accordingly.
> >
> > I found the clarification of the modelling of $-\sigma^2$ very interesting and would think it would be a further useful contribution of the paper to include an ablation study on this in the appendix of the final paper as it is contrary to the current standard practice in the literature.

---

### Author Rebuttal · Authors · 2023-08-10


We'd like to thank all reviewers for your insightful comments, and for acknowledging our contributions.

Specifically, we are encouraged to hear that our methods are "easy to implement" (Reviewer QJyu and Reviewer 2mxf) and hence practical, and that our contributions are "of interest to the Bayesian optimization research community as well as to practitioners" (Reviewer 4YBP and Reviewer 2mxf). Regarding our theoretical contributions, Reviewer 2mxf has commented that **"our technical contributions are solid and non-trivial"**, and Reviewer VcKK has mentioned that **"the theoretical analysis of the algorithms is very strong"**. Our empirical contributions have also been acknowledged, for example, Reviewer VcKK has commented that **"papers like this one take a big step towards making BO more practical to the wider scientific community"**.

Here we provide some important additional experimental results, and we'll address your individual questions in our separate responses below.

---

A common suggestion to further improve the empirical contributions of our paper is to add **real-world experiments with higher-dimensional continuous input spaces**. Therefore, following your suggestion, we have added two real-world experiments with 12-dimensional and 14-dimensional continuous input spaces, respectively:

- Lunar landing experiment: Here we tune $d=12$ parameters of a controller which is used to control a lunar lander in the OpenAI gym environment, in order to maximize the cumulative rewards in an episode.
- Robot pushing experiment: Here we tune $d=14$ parameters controlling a robot, in order to make it complete a task involving pushing objects. The goal is also to maximize the cumulative rewards.

Both experiments are commonly used benchmarks in the literature of high-dimensional Bayesian optimization [13]. More importantly, for every evaluated set of controller parameters in both experiments (i.e., every input $x$), the observation (i.e., cumulative rewards) is noisy due to random environmental factors. In addition, the noise may be heteroscedastic. For example, an effective set of parameters which can reliably and consistently control the robot is likely to induce small noise variance, whereas some ineffective sets of parameters may cause radically varying behaviors  and hence large noise variances. Therefore, these experiments are also suitable for the application of our algorithms.

The results are shown in Fig. 1 in the attached pdf, which demonstrate that **in these experiments with high-dimensional continuous input spaces, our algorithms still consistently achieve compelling performances** (especially with  𝜅=0.3 , which is consistent with our original experiments, see lines 258-261).

We think that these additional results serve as further evidence for the empirical effectiveness and robustness of our algorithms. We sincerely hope these results, together with our individual responses below, could improve your assessments of our paper.

---

### Decision · Program_Chairs · 2023-09-21

**Decision:**

Accept (poster)

**Comment:**

This manuscript considers the problem of Bayesian optimization for batch experimental design. The authors focus on the question of replications, where the experimenter may wish to make multiple independent measurements for the same design in order to reduce uncertainty of the resulting estimate. In the presence of heteroskedastic noise, we might wish to use a variable/adaptive number of replications in different regions of the design space, which raises interesting questions regarding experimental design. The authors propose a novel algorithm for this setting and provide both theoretical convergence guarantees and a empirical investigation into its performance.

Overall, the reviewers agreed that the proposed algorithm, albeit simple, is well motivated and a valuable contribution that will be of interest to the NeurIPS community. The reviewers also generally agreed that the empirical study was well designed and that its findings provided convincing evidence of the algorithm's real-world performance.

I encourage the authors to reflect upon the reviews and discussion in updating their manuscript.